# Detecting Respiratory Viruses Using a Portable NIR Spectrometer—A Preliminary Exploration with a Data Driven Approach

**DOI:** 10.3390/s24010308

**Published:** 2024-01-04

**Authors:** Jian-Dong Huang, Hui Wang, Ultan Power, James A. McLaughlin, Chris Nugent, Enayetur Rahman, Judit Barabas, Paul Maguire

**Affiliations:** 1School of Computing, Ulster University, Belfast BT15 1AP, UK; 2Wellcome Wolfson Institute for Experimental Medicine, School of Medicine, Dentistry and Biomedical Sciences, Queen’s University Belfast, Belfast BT9 7BL, UK; 3NIBEC Nanotechnology & Integrated Bio-Engineering Centre, School of Engineering, Ulster University, Belfast BT15 1AP, UK

**Keywords:** Near-Infrared (NIR) spectroscopy, portable, handhold, respiratory syncytial virus (RSV), Sendai virus (SEV), detection, Artificial Intelligence, machine learning, variable importance in projection (VIP) scores, Quantile, variable truncation

## Abstract

Respiratory viruses’ detection is vitally important in coping with pandemics such as COVID-19. Conventional methods typically require laboratory-based, high-cost equipment. An emerging alternative method is Near-Infrared (NIR) spectroscopy, especially a portable one of the type that has the benefits of low cost, portability, rapidity, ease of use, and mass deployability in both clinical and field settings. One obstacle to its effective application lies in its common limitations, which include relatively low specificity and general quality. Characteristically, the spectra curves show an interweaving feature for the virus-present and virus-absent samples. This then provokes the idea of using machine learning methods to overcome the difficulty. While a subsequent obstacle coincides with the fact that a direct deployment of the machine learning approaches leads to inadequate accuracy of the modelling results. This paper presents a data-driven study on the detection of two common respiratory viruses, the respiratory syncytial virus (RSV) and the Sendai virus (SEV), using a portable NIR spectrometer supported by a machine learning solution enhanced by an algorithm of variable selection via the Variable Importance in Projection (VIP) scores and its Quantile value, along with variable truncation processing, to overcome the obstacles to a certain extent. We conducted extensive experiments with the aid of the specifically developed algorithm of variable selection, using a total of four datasets, achieving classification accuracy of: (1) 0.88, 0.94, and 0.93 for RSV, SEV, and RSV + SEV, respectively, averaged over multiple runs, for the neural network modelling of taking in turn 3 sessions of data for training and the remaining one session of an ‘unknown’ dataset for testing. (2) the average accuracy of 0.94 (RSV), 0.97 (SEV), and 0.97 (RSV + SEV) for model validation and 0.90 (RSV), 0.93 (SEV), and 0.91 (RSV + SEV) for model testing, using two of the datasets for model training, one for model validation and the other for model testing. These results demonstrate the feasibility of using portable NIR spectroscopy coupled with machine learning to detect respiratory viruses with good accuracy, and the approach could be a viable solution for population screening.

## 1. Introduction

Studies for the development of respiratory virus detection methods, especially those with low cost, rapid ease of use, and high accuracy, are in urgent demand under the circumstances of the pandemic of novel coronavirus disease 2019 (COVID-19), caused by the severe acute respiratory syndrome coronavirus 2 (SARS-CoV-2) infection, which has brought about a huge loss of human life and economy [1,2,3]. The situation may even deteriorate as various virus mutations have emerged [1,4]. Conventional methods of virus identification or virus infection detection need certified laboratory and trained professionals to carry out, at relatively high cost, and are time-consuming for specimen transport in many situations, processing, and test report generation. This is the case for RT-PCR (Reverse Transcription Polymerase Chain Reaction) [5,6,7], the gold standard method. Serological immunological assays, covering Enzyme-Linked Immunosorbent Assays (ELISA) and Colloid Gold-based Immunochromatographic Assays (CGIA) [8,9], aside from physical means such as chest computed tomography (CT) [10,11,12], are further instances.

Optical methods have been applied to virus detection and viral infection detection. Fernandes et al. used near infrared (NIR) spectroscopy for rapid detection of Zika virus [13], San-Blas et al. applied mid infrared spectroscopy (MIR) for detection of Meloidogyne infestations [14], and Zhang et al. reported use of Surface Enhanced Raman Scattering (SERS) in interrogation of SARS-CoV-2 [15]. In general, reports of NIR spectroscopy to directly detect respiratory viruses, especially SARS-CoV-2, have been rare in the literature. Cases of cognate work, e.g., [16,17], are of a different theme from the study here.

A portable NIR spectrometer, being an optical instrument, has its advantages regarding rapidity, portability, and low cost and, if successful, is a strong contender for widespread deployment for rapid virus detection. Application of this method to detect SARS-CoV-2 or the effect of the viral infection has rarely been reported in the literature, partly due to the relatively lower quality and inadequate specificity and accuracy of such instruments. Artificial intelligence (AI), including machine learning approaches, may provide mitigation of these drawbacks, and then the application of such techniques to identify features in portable NIR spectra of virus samples would be invaluable. This study is therefore designed to use a portable NIR spectrometer to detect the presence of respiratory viruses, underpinned by AI approaches. As a first phase of this study, two common respiratory viruses, the Respiratory Syncytial Virus (RSV) [18] and the Sendai Virus (SEV) [19], have been used as the analytes dispersed in phosphate buffered saline (PBS) in order to optimise the protocols with the relatively less harmful viruses in laboratory conditions, in preparation for the later phase to investigate detecting the more dangerous SARS-CoV-2 or even its mutation viruses. The primary objective of this data-driven study is to examine the possibility and feasibility of detecting the presence or absence of a respiratory virus, instanced by the RSV and SEV, in an in vitro, aqueous sample using a portable NIR spectrometer coupled with machine learning.

Summing up the above, to cope with the large-scale respiratory infectious disease like the pandemic of COVID-19 that threatens human health, methodology for detecting the viruses is essential; the lab-based benchtop instrument methods, covering RT PCR, ELISA, CGIA, and SERS, etc., have their inadequacy in terms of cost, potability, ease of use, especially the rapid mass deployment; NIR methods include lab-based benchtop ones and portable ones, with the former having similar shortcomings as those mentioned above, while our study is exclusively focused on the latter, for taking advantage of the latter’s merits and avoiding the former’s shortcomings.

However, the spectra from a portable NIR spectrometer demonstrated interweaved curves of the virus-absent and the virus-present samples, representing an obstacle in distinguishing between the two as a feature of the relatively poor quality of the instrument, leading to a possibility of failure in the detection effort with this type of instrument. AI techniques, as applied broadly in many research fields, were then deployed to try to solve the problem. There is a large amount of literature to underpin the validity of the application, such as [20] (R. Hecht-Nielsen) that demonstrates the capacity and reliability of neural networks to approximate real-world objects (or curves) and indicates that “any L2 function can be implemented to any desired degree of accuracy with a three-layer backpropagation neural network” (note: A function that, over a finite range, has a finite number of discontinuities is an L2 function [21]). This means that no matter how complex a curve is, there is a proper way to approximate it through AI approaches. The fundamental works like this formed the basis of the belief and confidence that the AI approaches would obtain trustworthy outcomes in solving the problem, as evidenced by our later results.

At the same time, our practice showed that using merely conventional machine learning algorithms with common hyperparameters to model the data resulted in relatively inadequate outputs in terms of accuracy that were around 0.7 to 0.8, as a reflection of the complexity of the study objects and the relatively poorer quality of the instrument, which was in general not as sensitive as expected. To overcome the hindrance, a VIP-Q algorithm combined with the spectra truncation method was introduced and applied in this study, with a promising outcome that was seen as a significant control factor leading to the encouraging results.

The hypothesis in this study can be stated as “Is it possible to detect viruses in solution with rapid and low-cost near-infrared spectroscopy so that the method can be used for effective screening and diagnosis?” Our work, with its exploratory nature, demonstrated such a vision that a positive answer to the hypothesis appeared to have been achieved with this study, or an affirmative answer to the hypothesis is shown to stand heretofore, and a way of solving the issue in this direction is hopeful. Consequently, we hope that this work, with its possible application prospects, will appeal to and encourage peers and the research/industry community to try further investigations and benefit society in the future.

## 2. Materials and Methods

### 2.1. Instrument and the Measurement System

The measurement system comprised an Ocean Optics NIRQuest512 NIR spectrometer with 512 pixels and a wavelength range of 899–1721 nm. It has an Ocean Insight HL-2000-HP-FHSA light source with a light output wavelength range of 360–2400 nm and a typical optical output power of 8.4 mW. The sample holder consists of a reflectance standard, a microscope glass slide, and a slide holder. A microscope glass slide is covered with parafilm coating, and a sample well is formed by removing the central 15 mm parafilm coating from the glass slide. The depth of the sample well is 120 μm. A drop (50 μL) of sample liquid was placed into the sample well of the glass slide and covered with a coverslip. NIR spectra were collected in reflection mode, illuminating the sample from above the cover slip. Figure 1 illustrates schematically the spectroscopic measurement system.

Reflectance data were obtained via the use of an Ocean Insight Spectralon diffuse reflectance standard, which provides reflectivity between 85 and 98% in the wavelength range of 800–2400 nm. Parameters used in the spectral sample data collection are: Integration time: 100 ms; Averaging: 5; Boxcar width: 5; Up to 50 spectra were obtained for each scan. The temperature of the laboratory was 21 ± 1 °C.

### 2.2. Sample Preparation and Spectra Data Collection

Two respiratory viruses, the human Respiratory Syncytial Virus (RSV) [18] and the Sendai Virus (SEV) [19], have been selected as the analytes for the spectral analysis, as they are typical respiratory viruses, relatively easily available, and relatively harmless to the laboratory operators. The former is a syncytial virus that causes respiratory tract infections, and the latter typically infects rodents and is not pathogenic for humans or domestic animals [18,19].

(1)Virus stocks (RSV titre: 2.8 × 10^7^ TCID_50_/_mL_; SEV titre: 10^8^ pfu/mL) kept at −80 °C were thawed quickly in a 37 °C water bath. All experiments used a different aliquot from the same stock.(2)Ten-fold serial dilutions of stocks were made with PBS—the RSV stock was diluted 10^−1^, 10^−2^, 10^−3^, and 10^−4^, while the SEV stock was diluted 10^−2^, 10^−3^, 10^−4^, 10^−5^, 10^−6^, 10^−7^, and 10^−8^.(3)For each dilution, a drop (50 μL) of virus sample was pipetted onto the sample well of the microscope glass slide and covered with a coverslip for measurement.(4)After each measurement, the microscope slide was washed in sterile water and dried with ethanol between each sample loading.

Sample NIR spectra data collection for different concentrations of RSV and SEV in the range 0.4–1.0 g/mL in PBS was obtained over four separate sessions. Four spectral datasets were collected on 15 October 2020, 29 October 2020, 20 November 2020, and 10 December 2020, named as datasets A, B, C, and D, respectively.

The four datasets (A to D) from different data collection sessions comprised a total of 2370 spectra containing RSV and SEV viruses in a PBS medium across all dilutions, along with samples containing only PBS medium. Those from samples containing virus(es) particles were designated as virus-presence (V1), and those without virus(es) particles were designated as virus-absence (V0).

### 2.3. Spectra Data Pre-Processing

Reflectance values were converted to absorbance, with a reflectance-to-pseudo-absorbance transformation formulated as A = log_10_(1/R) [22,23]. The data pre-processing steps [24,25,26,27,28,29,30] were carried out on the spectral data:(1)Smoothing (filtering) [24]: The absorbance spectra were subjected to a Savitzky-Golay (SG) filter, which reduces random noise while preserving useful spectral information. It was processed with order 2 and window length 13.(2)Multiplicative Scatter Correction (MSC) [29,30]: To correct baseline distortion, which is mainly a result of light scattering [27].(3)Standard Normal Variate (SNV) [26,27,30]: To reduce the scaling differences between sample spectra. For a given spectrum, SNV subtracts the mean value of the spectrum from each variable. The results are divided by the standard deviation of the spectrum.

A combination of pre-processing methods was used for all the datasets, named pp1 to pp6 (Table 1).

Combinations of the above pre-processing steps were tested, and the best-performing combinations for a specific spectral set were identified and used as a data processing pipeline for data modelling. The spectra, after undergoing a variety of different pre-processing, have been plotted in figures in Appendix A, Appendix B and Appendix C. Figures in Appendix A are discussed as examples and presented in Section 3.1 of the main manuscript.

Noticing that literature has shown that the MSC and SNV have highly correlated correlations (e.g., [30]), pp6 as a sequential processing after pp5 might be seen as duplicated. But this was still tested and presented as a case study example, which may further demonstrate the conclusions in the literature with the pp6-related output figures. On the other hand, the performance of the models from the datasets generated by pp6 appeared relatively better than that of pp5 (Section 4). This is interesting and worthy of further investigation.

### 2.4. Variable Selection: A Special Algorithm with Quantile Lines for the VIP Score Curves and the Data Series Truncation Treatment

#### 2.4.1. VIP (Variable Importance in Projection) Score

Variable selection plays an important role in building high-quality AI models by reducing the dimension of the data space and identifying informative variables. In this study, Variable Importance in Projection (VIP) scores [31,32,33] have been used as a measure of a variable’s importance to the model outcome. The generation of the VIP method is closely related to PLS regression and PLS-DA models [31,34]. In the PLS literature, it is defined for each of the variables, and the VIP score of a variable is a summary of the importance of the projections to find h latent variables (here h is smaller than the total number of predictors p) [35,36]. VIPs identify the most relevant variables related to a dependent Y in an independent array X, and a variable with a higher value of the VIP score shows that it is more relevant to predict the response variable [31,36]. VIP finds its broader application in further development of NIR data analysis (e.g., [37,38]), and with the same object to deal with, the usage of this approach was extended in our study to serve variable selection in machine learning approaches. It resulted in dimension-reduced and more informative variable subsets for the SVM, PLSDA, kNN, and NN approaches in this study, leading to a significant improvement in the output.

This measure helps identify regions in the wavelength spectrum that contribute most to differentiating different classes, as illustrated in Figure 2 and Figure 3. The higher the VIP score of a wavelength, the more contribution it makes to the classification. Therefore, this measure provides a way to identify those important variables (wavelengths) in terms of their contribution to identification.

A full set of the VIP scores for RSV and SEV, with six different pre-processing combinations, is given in Appendix D.

#### 2.4.2. Quantile Value as a Threshold Line

After calculating the VIP score for each of the wavelength variable points, those points with their VIP scores above a given threshold are selected to form a subset of the whole variable set, containing relatively more important variables. This subset is then used as a feed-in set for machine learning modelling. The threshold line, with its value being a quantile (Q value) of the whole VIP score set, varies as assigned in a small step from low to high or reversely, forming a series of the variable subsets. Among them, the one that yields the best model output with the highest accuracy is identified and finally selected, and the generated model with this subset is adopted to be the optimised and the best one under the Q value. This processing plays a role in variable selection. It can also be seen as a pre-training procedure.

In this study, firstly, the VIP scores were calculated for each of the wavelengths involved, and the VIP scores versus the wavelengths were plotted (an example is shown in Figure 2). Then, a threshold value Q was assigned (an example is shown in Figure 3) for the selection of features (wavelengths) to be included in a subset of the dataset for further modelling. Q is a quantile value of the full set of VIP scores for a spectrum dataset; its upper limit is the maximum of the VIP scores in the full set.

As an illustration of the algorithm for this VIP score feature selection, in Figure 3, the VIP score curve for a given spectrum is presented. A y = VIP (Q) line with Q = 0.4 separates the VIP score points into upper and lower regions. Q = 0.4 means that 40% of all the VIP score points in this dataset are lower than the y = VIP(Q) line, and the rest are higher than the line. These high VIP score points define a series of subintervals in the wavelength axis, as shown by the vertical lines that form shaded regions. On the corresponding spectra diagram from which this VIP score plot is yielded, the spectra within these subintervals of wavelength then form a subset of the full spectra dataset. Carrying out the variable selection procedure, this subset is first sought and formed from the training dataset (full set); then, for correspondence between the training and the validation and/or testing datasets, the same Q is used on processing the validation and/or testing dataset (full set) to form a corresponding subset with the same number of wavelength points as in the training subset. Thus, two or three subsets, a training one plus a variation one (if required) and a testing one, are generated for further modelling. The sub-datasets contain only variables with a higher contribution to the classification, and thus a better modelling outcome is expected.

The Q is a statistical concept. An advantage of using this quantile value is to avoid dealing with various VIP score values that vary for different spectral sets and to provide a unified metric for processing and comparing. Note that the VIP score value corresponding to the Q value of 0.4 is 0.68 in Figure 3, different from each other. But a quantile value Q is applicable to compare with one from any other spectrum set and is easy to use.

In the modelling experiments, Q values were assigned, e.g., from 0.1 to 0.9 with step 0.1. The corresponding outputs were compared, and those with better accuracy were identified and selected as the optimised outputs.

#### 2.4.3. Find an Optimised Q and Associated Model

Under a fixed Q and its associated subset, certain number (10 times in this study) of modellings were carried out. The output of the runs, metrically represented as Accuracy, were averaged to a mean accuracy that was attached to this Q. Varying the Q from 0 to 1 with a small step (which was 0.1 in this study), a series of mean accuracies were obtained relevant to their Qs. The maximum one of the mean accuracies was then identified, corresponding to the highest-accuracy-yielding Q and the generated model. This model was seen as the best one produced and could be applied for further classification prediction on an unknown dataset, as long as its accuracy met some criterion, e.g., 0.9. In this treatment, the finally identified Q is an indicator for the model that has the best accuracy.

Figure 4 illustrates that the mean accuracy output (Y) from trained models varies with given Q values (X). A good accuracy value of 91.99% at Q = 0.13 in Figure 4a and two above 93% at Q = 0.1 and 0.9, respectively, in Figure 4b, can be identified. Here, the number of repeated modellings was 10. If the number of repeat runs could be larger, the results could be more representative and convincing. But the computation resources restricted the use of a very small step to go through the interval of Q for a more holistic checking for a ‘best’ Q value. As an alternative, a smaller step was set for within the neighborhoods of a ‘coarsely’ determined Q value, as was the case in Figure 4a for a step = 0.01 around the Q value of 0.1, which was identified as yielding a high accuracy, in the neighbourhood [0, 0.25]. Thus, in general, limited additional runs can be arranged for Q values within this limited subinterval to further establish a better Q, corresponding to an optimised model with relatively high accuracy, without trying all the Q values in [0, 1]. This is a compromised way to establish a high-accuracy model without having adequate computing resources. On the other hand, a too small step may not make sense as it would not vary the number of variable points selected when the difference made by the step change is smaller than the minimum difference of two adjacent spectra points.

Figure 5 is an illustration of the processing for selecting a sub-dataset for training and a corresponding one for testing through VIP scoring under a given Q value.

#### 2.4.4. The Truncation Process

Figure 6 shows the curves as the difference between the preprocessed and un-preprocessed (original) spectra, or the removed component from the spectra by the filtering and smoothing processing, which at least partly contain the noise of the spectra. As curves abruptly increase their fluctuation amplitude in the region close to the right end of the wavelength range, which might reflect the occurrence of extra noise in that region, the processing of the dataset with wavelength points truncated from the right end side was deployed to try to examine the performance of the modelling without being affected by the possible extra noise. The original wavelength points were 1 to 512 (corresponding to the wavelength of 899 to 1721 nm), and for a trimmed dataset, they were 1 to 450 (corresponding to the wavelength of 899 to 1622 nm).

The method presented and described is for further use of the identified features (variables or wavelength points) to form a more proper subset of the dataset for better modelling output. This special variable selection algorithm, with its adequate performance demonstrated in later sections, provides a solution to improve the quality of the yielded AI models and is expected to play a significant role in the similar study of NIR spectroscopy.

### 2.5. AI Algorithms

The relationship between pre-processed data and virus label information was modelled using approaches of SVM (Support vector machines), kNN (k-nearest neighbours), NN (Neural Network), and PLS-DA (Partial Least Squares Discriminant Analysis), which are common methods in machine learning and thus adopted in this study for the modelling. An SVM is a binary linear classifier with a non-linear step called the kernel transformation [39]. A kernel function transforms the input spectral space into a feature space by applying a mathematical transformation (often non-linear). Then, a linear decision boundary is fit between the closest samples to the border of each class (called support vectors), thus defining the classification rule. KNN is a non-parametric classification method used for classification and regression. In both cases, the input consists of the *k* closest training examples in the dataset. The kNN classification is used to classify unlabelled observations by assigning them to the class of the most similar labelled examples. The input consists of the k closest training examples in the feature space [39,40,41,42,43]. An artificial neural network (NN) is a mathematical or computational model that tries to simulate the structure and functional aspects of biological neural networks. It consists of an interconnected group of artificial neurons and processes information with a connectionist approach to computation. The properties of the network are determined by its topology and the properties of the neurons [43,44,45,46,47,48].

PLS-DA (Partial Least Squares-Discriminant Analysis) can be regarded as a linear classifier. The method (or a method that includes PLS-DA as one of its steps) aims to find a leaner function that divides the space into different regions. Fundamentally, PLS-DA predictive modelling encompasses two main procedures: (a) dimension reduction and (b) prediction model construction (i.e., discriminant analysis), which are combined into one algorithm and are especially applicable to modelling high-dimensional (HD) data [49,50,51,52,53].

### 2.6. Data Modelling: Experiment Design and Protocols

Models of classification were constructed to identify the samples of virus-presence (V1) versus virus-absence (V0). Two protocols were developed and implemented in spectral data modelling, to investigate the performance of models under different conditions for algorithm selection and find optimised ones for virus detection.

*Protocol I:* Using 3 out of the 4 datasets for training and the rest for testing, i.e., the training set comprised 3 datasets out of the 4, and the other 1 being the test set. This means the dataset A, B, C, and D in turn was used as the testing set, and the corresponding rest 3 were merged as the training set. There were 4 combinations of the training/testing pairs: aggregation of datasets B, C, and D (training) versus dataset A (testing), then (C + D + A) versus B, and (D + A + B) versus C, plus (A + B + C) versus D. This approach therefore explores the impact of the inter-session data mixture on model output or accuracy, which may reflect more realistically a future application environment. Under the protocol, every combination of the spectra datasets subjected to each of the 6 types of pre-processing (pp1 to pp6) was taken in turn as the input of a model. Thus, for a full set of the experiment, there were 4 × 6 = 24 sets of trials of modelling (see Section 3.2.1).

*Protocol II:* For the 4 spectra data sets A, B, C, and D, merge 2 of them for training, 1 for validation, and the other 1 for testing. For example, A + B is for training, C is for validation, and D is for testing. There were a total of six combinations of the four datasets considered for a full set of trials (see Section 3.2.1). Aside from *Protocol I*, *Protocol II* was adopted not only for algorithm selection and model testing but also for considering the necessary model validation under the conditions of this study.

The protocol design considered carrying out the validation and testing in special ways for the special datasets in this study. *Protocol I* contained only model training and the test process, but *Protocol II* involved training, validation, and test phases.

Metrics for evaluating the model output are listed in Table 2.

### 2.7. Model Training, Selection, and Validation

The whole procedure can be summarised as follows:Remove the outliner and high noise signal points for the original datasets and perform the truncation treatment to form a datasets-group1; pre-process with pp1 to pp6 on this group1, to obtain a dataset-group2; find and attach VIP scores for each of the datasets in dataset-group2; search and find a good Q value: for each of the Q values varied in a small step in the interval (0, 1), find a sub-dataset with the VIP score above the one corresponding to the Q value to form a sub-dataset, all of which then form a sub-dataset group3 (Figure 3, Figure 4 and Figure 5).Using each dataset in sub-dataset-group3, carry out the modelling with SVM, PLSDA, kNN, and NN. Approximately 10 runs were conducted for each of the algorithms and conditions, and the average of the outputs was adopted as the final output. For SVM, ‘Kernel Function’ was set as ‘linear’, and ‘Kernel Scale’ was set as ‘auto’, with the MATLAB 2021a toolbox. For PLSDA, the first latent variable was used, with its value being the indicator to calculate the accuracy and evaluate the model performance. For kNN, default parameters were adopted from the MATLAB 2021a toolbox; ‘Optimize Hyperparameters’ and ‘Hyperparameter Optimization Options’ were adopted for optimised modelling. For NN, an BP-MLP (Back Propagation Multi-Layer Perceptron) neural network model structured with 5 layers (1 input, 1 output, and 3 hidden) and 200 neurons per layer was adopted after trial runs. For other parameters, MATLAB default values were adopted.Their output was used to calculate the accuracy parameter for performance evaluation (tables in Section 3). A set of ‘qualified’ models with their good accuracies were selected, with the specific PP and Q adhered to each specific model.The above modelling procedures were conducted with *Protocol I* and *Protocol II*. Accuracy was used as the main metric to evaluate the output of the modelling, for algorithm/model validation and test practice.

The modelling was performed on a personal computer with a configuration of an i7 processor@1.8 GHz/1.99 GHz and 16 GB of memory. All models are implemented in MATLAB 2021a. For each experiment, 10-time runs were performed for *Protocol I*, and 3-time runs were applied for *Protocol II*, in consideration of both the soft computing nature and the computation resources.

## 3. Results

### 3.1. Spectra Features and VIP Score Plot

Spectra were plotted in three categories: (1) spectral curves for the datasets that had undergone the six types of pre-processing; (2) comparison of the mean spectrum of samples with virus(es), termed V1 (virus-presence), versus the mean spectrum of samples without virus(es), termed V0 (virus-absence).

For all the sessions’ data, having undergone 6 types of pre-processing (pp1 to pp6), respectively, the spectra were plotted for visualisation and comparison, presented in the Appendix A, Appendix B, Appendix C and Appendix D as the spectra of virus-presence (V1) and virus-absence (V0) samples:Appendix A—RSV,Appendix B—SEV,Appendix C—RSV + SEV,Appendix D—VIP scores. RSV, SEV, RSV + SEV, under pp1 to pp6

In the above list, RSV + SEV means a dataset with both RSV spectra and SEV spectra of a session, but not that of the spectra of a physical mixture of the two, which, after the presented trial, shall be a scenario in experiments designed in the future to investigate the possibility and feasibility of detecting multi-viruses in a sample.

Figure 7a–f (reproduced as Appendix A, Figure A1, Figure A2, Figure A3, Figure A4, Figure A5 and Figure A6) shows the spectra of virus-presence(V1) and virus-absence(V0) RSV samples, having undergone different pre-processing (pp). Figure 7a shows the raw spectral data, as %Reflectance, smoothed with the Savitzky-Golay algorithm (pp1). The process of normalisation (SNV) results in a shifting of the spectral intensity response to positive and negative values (Figure 7b, pp2). In Figure 7c, the (SG) smoothed Reflectance values have been converted to Absorbance (pp3), while in Figure 7d, the spectra have been additionally normalised by SNV (pp4). In Figure 7e, the MSC algorithm has been applied to the smoothed (SG) Absorbance spectra (pp5), while in Figure 7f, these spectra have been further subjected to SNV (pp6) (see note in Section 2.3). It can be seen that the process of SNV significantly reduces the inter-spectral variability, as does the process of MSC. Similar results are observed for the spectral responses of the SNV-processed samples, as shown in Appendix A, Appendix B and Appendix C.

It is obvious that the curves are interweaved in the Figures and that it is difficult to distinguish visually between the spectra of virus-present and virus-absent ones. This represents an obstacle to detecting viruses with this type of spectroscopic instrument. Correspondingly, the machine learning approaches in our study provide a necessary and significant means of coping with the difficulty.

As general features of the spectra, there are peaks at around 980 nm, 1150 nm, 1450 nm, and 1640 nm, with the strongest absorption occurring at ~1450 nm. The weak and medium-strength broad features appear to be close to those of the NIR spectra of water, resulting from overtone (2𝜈_1_ and 2𝜈_3_) and combination (𝜈_1_ + 𝜈_3_) bands [54,55].

In these figures, the curves of mean V1 and mean V0 are very similar. Any subtle differences in the respective V1 versus V0 spectra are visualised by either the ratio of the mean Reflectance (pp1, pp2) or difference of the mean Absorbance (pp3 to pp6) spectra. Figure 8 shows the examples for RSV spectra. In all cases, in the absence of normalisation (SNV), the ratio/difference spectra indicate that the primary difference between the spectral datasets are a reduction in the overall strength of the water features. A similar behaviour has previously been reported for the spectra of analytes in aqueous solutions, for example, urea [56]. Having undergone normalisation, the ratio/difference spectra consistently indicate the presence of a differentiating feature centred at ~1450 nm. The relationship between this feature and the presence/absence of the virus needs further investigation.

Therefore, although the spectra of the virus containing PBS solutions were dominated by the absorption of the PBS itself, the presence of the virus resulted in subtle but measurable changes to that spectrum, and those changes were detectable for each virus, suggesting that solutions containing each can be identified and differentiated.

Figure 9 shows the difference (meanV1 vs. meanV0) of the 4 data sessions for RSV with the pre-processing methods pp3 and pp4. Aside from similarly appearing a peak in the interval [1400, 1500], there was poor reproducibility of the pattern of the curve.

In general, the problematic reproducibility of (V1–V0) in different sessions and the small magnitude of the difference (V1–V0) may trigger a concern (in a real implementation sense) about using the visualisation means to detect the analytes, which can represent a negative factor and a serious challenge in the detection attempt. But on the other hand, the existence of the features does exhibit a firm physical ground for the realisation of the detection. Using other means that are both able to have a good effect on the detection of the different spectra and avoid doubtable queries, such as AI approaches, is a realistic solution. The visualisation and the machine learning means can play a role in mutual confirmation and corroboration for reaching this study objectives, with their interdependence.

It appears that with this low-cost spectrometer, it is very difficult to obtain reproducibility in terms of even a stable feature such as the orientation of the curves, let alone the biomarker. This reflects the functioning limitations of the instrument, likely determined by its intrinsic property as a low-cost one. Its limited capacity leads to frustration when comparing its functioning ability with that of a lab-based, sophisticated instrument, which is consistent with its internal nature. Therefore, the notion that it is difficult and unfeasible to use a low-cost portable spectrometer to find out a clear biomarker has received one more endorsement from the output of this case study. Meanwhile, the demand for using remediable means to help realise detection with good output quality is more prominent and imperative when adopting the portable spectrometer.

Figure 10 and Figure 11 are the output of the PCA (Principal Component Analysis) for the RSV and SEV samples of all data sessions, which show their first and second primary components, respectively, indicating that the analytes appear to be separatable and detectable, although with some degree of difficulty, and providing a ground for the AI approaches to work on.

### 3.2. Modelling Output

#### 3.2.1. Results of Modelling with *Protocol I*

It will be more practical and necessary to classify spectra in an unknown session with a trained model for real world applications. From an application perspective, a trained model will be used to predict the results of unknown samples (measured under the same data collection protocol). Starting from this point, trials were attempted using three-sets for training and the ‘unknown’ one-set for testing (Section 2.6) to predict the classification of the test set. Not surprisingly, the results with conventional machine learning methods on the original spectra data using common hyperparameters were poor; the accuracy in a V1 vs. V0 classification was averaged at 0.50 for the RSV and 0.78 for the SEV datasets. Thus, the VIP score–Quantile approach plus variable truncation processing were applied to overcome the obstacle and improve the output.

With the aid of the Q value selection procedure, models were constructed for V1 vs. V0. Table 3 contains the results of the treatments mentioned above, with and without the truncation, for the approaches of SVM, PLSDA, kNN, and NN, respectively.

For a Q value, modelling was performed for combinations of pp1 to pp6 on the dataset A, B, C, and D as the test dataset, respectively, 24 in total, which was designated as one run. To express and evaluate the modelling output, symbols and metrics have been defined as follows:

(1). A (i, j): the accuracy values obtained in one run, which is a matrix of 6 rows (representing pp1 to pp6) and 4 columns (representing the situation of using test sets A, B, C, and D), i.e., i = 1 to 6 and j = 1 to 4.

(2). AccPPmax(j): Maximum of the accuracy in a group of pp1 to pp6 output, for a fixed j (j = 1 to 4, corresponding to the test set being A, B, C, and D, respectively),

AccPPmax(j) =$\underset{1\leq \;i\;\leq 6}{\max}A\left(i,j\right)$, j = 1, 2, 3, 4.

(3). Mean_AccPPmax:

Mean_AccPPmax = average [AccPPmax(j)) $=[\overset{4}{\underset{j=1}{\sum }}\mathrm{AccPPmax}\left(j\right)]$/4

The maximum [A (i, j)] from (1), the maximum of Mean_AccPPmax from (3), and the average of Mean_AccPPmax for 10 runs have been used as the items (i), (ii), and (iii), respectively, in Table 3a–d.

The prediction accuracies shown in Table 3a–d are a consequence of the VIP score–Q algorithm plus the truncation treatment. It can be seen from the Tables that the algorithm NN performed the best in the classification. Also, among the analytes RSV, SEV, and RSV + SEV, SEV tends to yield an outcome superior to the others.

#### 3.2.2. Results of Modelling with *Protocol II*

Results of modelling with *Protocol II* are shown in Table 4a,b, for the algorithms SVM, PLSDA, kNN, and NN, respectively. Average values of the accuracy for different combinations of the training/validation/test groups have been calculated and listed at the bottom of each table to give a general view of the performance of the modelling.

Modelling outputs were listed in the tables for each combination of the datasets A, B, C, and D assigned as training, validation, and test datasets. For example, within Table 4b, the rows with their first column as ‘A + C’ give the accuracies as the output of kNN and NN modelling; for the NN output, with A + C merged as the training dataset, B as the validation set, and D as the test set, the accuracy for RSV, SEV, and RSV + SEV was 1.00, 1.00, and 1.00, respectively, for the validation, and 0.97, 0.99, and 0.94, respectively, for the test of the models underwent validation, respectively. The averaged accuracy in Table 4 is summarised in Figure 12, which demonstrates the validation and test outcomes of the algorithms and the basis of good model selection.

In general, among the four algorithms deployed in the trial, NN outperformed others with its best accuracies all above 0.90 in establishing the multivariable models, similar to the outcome of *Protocol I*; and also, SEV tends to receive the highest scores among the others. Eventually, NN appears to be the most optimal of the algorithms tried. At the same time, the optimised models corresponded to their associated Q values as part of the model output.

With the assigned datasets, the validation process provides initial testing against unseen data, then evaluates how well the trained model makes the prediction based on the fresh data. The protocols used in this study involved both types of processing, i.e., training-test and training–validation-test. In implementing *Protocol II*, especially for the outperformed algorithm NN, with the validation output (the accuracy) all better than 0.90, no hyperparameters were adjusted for the models created. Thus, the validation and test processing can be seen as two independent tests with an adequate outcome.

## 4. Discussion

### 4.1. On the Sample Preparation

The logic of the experimental design was to first determine if AI and machine learning algorithms can or cannot be used to detect the presence of several different viruses in a solution using portable NIR data. Our initial aim was also to use “pure” and relatively strong virus portable NIR signals to train AI. Our further ongoing studies are building on such baseline data to distinguish possibly multiple different viral signals or weaker signals from more noisy samples. PBS is a simple, inexpensive solution that can be used to stabilise and neutralise pH levels in biological samples and could very well be used for this purpose when collecting viral samples (from swabs). The downside of PBS in this case is that while it preserves viral structures, it may also keep any viruses in an infectious state. While this could be a problem in terms of clinical practice and the handling of human samples that may still be infectious, it is also very advantageous for preserving the NIR signal. The issue of infectious samples can be addressed by post-measurement inactivation of samples by several different methods (addition of lysis buffer, heating).

RT-qPCR is usually considered the most sensitive and selective method to detect the presence of virus genomes. The best-in-class assays can detect ~100 copies of viral RNA/mL for SARS-CoV-2, although currently accepted assays have LoDs that may vary over 10,000-fold. This method does not differentiate between virus particles that can infect and those that cannot; it requires several sample preparation steps; it is expensive; and it requires large and expensive laboratory equipment and expert analysis.

A portable NIR-based detection method is aimed at working with minimal to no sample preparation, being field-portable, inexpensive, and user-friendly with automated analysis. The aim of developing such a system is not to create one that is a competitor, i.e., better in detection parameters than the gold standard qPCR, but rather to create one that has a detection range relevant for rapidly detecting pathogens in patient samples in the field, at a low operating cost and good scalability. The limits of detection of respiratory viruses using our NIR technology remain to be determined.

When establishing new techniques or experiments with viruses, it is best practice in virology to prepare a single stock of virus that is aliquoted and stored for repeated use during the protocol validation. Virus stocks are crude preparations derived from infected cells, and each stock preparation will differ to some degree in terms of the quantity of infectious and non-infectious virions. As such, in order to validate the ability of the technology to repeatedly detect individual virus species and distinguish between different virus species, it is imperative to use the same stocks of virus repeatedly. A single stock of each virus species that is aliquoted and frozen allows for several independent experiments to be undertaken with the exact same material, which is essential to clearly demonstrate the capacity of the technology to repeatedly distinguish between the presence and absence of viruses under investigation. The next phase of this study would be to translate this into appropriate clinical samples.

We recorded up to 50 spectra per sample with the following parameters (under Section 2.1): averaging: 5, boxcar: 5, integration time: 100 ms. These up to 50 spectra were replicated measurements of the same sample. The spectroscopic device gives 50 or so spectra in the measurement. Therefore, we used all the spectra from some sessions to build a model for it to be tested on spectra from other sessions, emulating a realistic use scenario.

This is the way the device gives its output, both in a laboratory and on site. We adopted using these spectra for model establishment, making use of all the data available, and at the same time to simulate the processing of data in a real-site application context, like our treatment of other aspects of this study, as long as it is possible.

### 4.2. About the Spectral Feature

The hardware of the spectrometer (Figure 1) contains a movable reflection probe with a holder (the black block) and a relatively fixed sample holder. In the data collection, the reflection probe with holder is put on the glass slide at the top of the sample holder, and then the scan starts and completes in about 2 min. The surface of the bottom of the prob block and the surface of the glass slide need to be contacted closely or ‘seamlessly’ in the scanning. But in real operation, the stability of the mounting is affected by so many factors and may not be ensured, e.g., the prob block is slightly deviated from its fitting position in the slot, resulting from the vibration of the surrounding environment factors or any other causes. Thus, a small gap may occur between the two surfaces. Note that the depth of the sample well is 120 μm and the relevant part is with such an order of magnitude as millimeter(s), any tiny deviation of the two surfaces from each other will enlarge the pathlength significantly than it really should be, which is one of the parameters to determine the effect of the light [22]. Consequently, the spectrum obtained can vary significantly from sample to sample. This is one of the possible reasons for the poor reproductivity observed in the spectra. Facing this issue, the machine learning approaches demonstrated their powerfulness and robustness in being able to ‘see through’ and penetrate the irregular features to catch the key information for detection and obtain results with relatively good accuracy.

Moreover, although the analysis of the collected data has exhibited a discrepancy between what it should be and what it really is in the reproducibility of finding (V1–V0), this can be seen as a simulation of a scenario that includes extreme rough working conditions that may be encountered when the detection approach is implemented for mass population screening in especially field settings with unstable environmental conditions and un-fully trained operators. Such a situation should necessarily be covered when considering the robust capability of the application.

The reproducibility problem is a concrete embodiment of the drawback of the portable instrument, and the AI approaches with the outcomes in Table 3 and Table 4 appear to have shown their ability to overcome the shortcoming.

In general, as it appears difficult to separate different classes from the visualised spectra (Figure 7; Figures in Appendix A, Appendix B and Appendix C), AI approaches play a key role in virus presence detection. The power of the AI approach is seen as embodied in its dealing with the irregularity and interwovenness of the spectrum, in particular. There are many factors that influence the variability of the output of the spectra scan, including the baseline variability and other variability that might occur, such as stretching along the wavelength axis or overall intensity variations, specific wavelength jitter, distortions, etc. One of the major sources of variability is the optical path, which will change every time the sample is changed. It could be expected that the 50 spectra taken with each sample would be very close together–since the conditions normally do not change much within the scanning time. For the next sample, the liquid height, etc., may be different, and the location of the glass slide with respect to the light source and optic fibre will be different. This could be affected by changed path distances, tilt angles, brightness, temperature, etc. Another source of variability that may appear on different days relates to the preparation of the virus samples themselves. It is difficult to exactly quantify and deal with the variability and its effects. At this point, the AI and machine learning approaches show their potential in coping with these by implicitly considering all those factors through their training procedures and processes and in distinguishing between the different classes of components. It is an interesting and attractive topic to try to identify the water peaks in the spectra and separate them from the possible net peaks of the virus itself but given this study objectives can be reached by bypassing this issue via AI/machine learning approaches, this can be an objective of another research effort.

### 4.3. On the VIP Score and Truncation Treatment

Apart from constructing an optimised subset of the training/testing data, VIP score distributions also provide important information to characterise the spectra of the virus investigated. Especially the VIP score curves reveal useful information in the identification of different classes (see the Appendix A, Appendix B, Appendix C and Appendix D).

The distribution of high VIP scores along the wavelength axis appeared some corresponding with that of the peaks of difference between V1 and V0 mentioned in Section 3.1 and in Figure 8 and Figure 9, suggesting that the position of the high scores is quite closely related to the NIR properties of water, incurring association of the relationship between the NIR spectra fingerprint of the viruses and the NIR properties of water that is worthy of further exploration. Concurrence with this feature, the high scores are not centralised in a special region but quite ‘broadly’ distributed in the wavelength axis especially when a Q’s value is not high.Significant wavelength positions match, to some extent, well for different pre-processing methods. This may suggest that the selection of a pre-processing method does not influence very much the result of establishing a good model, and then the pre-processing procedure may be simplified by using fewer pre-processing method(s).Experiments show that better output with higher accuracy does not necessarily result from the sub-datasets corresponding to a small number of variable points trimmed from higher VIP scores or Q lines. Test runs for the quantile value from 0.9 down to 0.1 and lower demonstrated that a lower Q value can lead to better output, as long as the Q value is above a small quantity. This may suggest that (a) VIP score–Q processing is different from conventional feature selection, which normally encourages selecting fewer variables (features) for building a sub-dataset to train the model. (b) Good practice via the VIP score–Q processing sometimes tends to select as many as possible variables (or useful information) to be enclosed into the subset to build the model, but excludes those wavelength points with very low VIP scores that are likely noise-impacted.

Meanwhile, the variable truncation process can improve, in many cases, the accuracy of the yielded models to some extent in terms of the item (iii) in Table 3a–d, i.e., the Average of Mean AccPPmax in 10 successive runs for a given Q. In general, the effect of this process on the improvement appears more significant for the kNN and NN that were the best-performing algorithms among the four deployed.

### 4.4. A Reflection of This Study

The purpose of this study is twofold. First, to find and confirm the existence of a model that can answer the questions raised by this study objective stated in the Introduction section—whether it is possible and feasible to use a portable NIR spectrometer to detect the presence of respiratory viruses. Second, to find an algorithm set that can generate models that are able to best serve the task of the first. The first question has been positively answered at the current stage in Table 3 and Table 4, i.e., the established specific model(s) can realise the classification or detection with a good degree of accuracy. The second is to develop an approach, including pre-processing, pretraining, and the AI algorithm(s), that are able to provide a general algorithm set that can generate good models to make classification predictions with satisfactory accuracy. The standard for the first is the existence of a good model, but for the second, it is a model generation system ensured by good probability. This is not ultimately completed when obtaining the results in the tables. It would be ideal if the model generation could be based on a larger number of repetitions and then be ensured better reliability in terms of a stable good probability of yielding good models. In the first task, 10 repetitions of the approach have been tried, and the results and their average values appear to be satisfactory, although based on the limited repetitions or model reproducibility. But advanced computational resources would make more repetitions feasible and yield a more robust outcome. Therefore, the results shall be enhanced if better computation resources are available for more repetitions of modelling further to the current progress, and the results listed in Table 3 and Table 4 are not necessarily the most optimal or the best models’ output.

### 4.5. Distribution of Good Accuracy over Different Pre-Processing Methods

Selecting a proper pre-processing method is a necessary concern when considering the real implementation of the algorithm set for a good model of detection. Figure 13 gives the distribution of good models with an output accuracy better than 0.9 over the six pre-processing (pp) methods, one for a 10-run output and the other for the output of three experiments each with 10 runs. It is apparent that for RSV, (pp3 and pp6) and (pp4 and pp6) are proper candidates for selection. For SEV, pp2 is obviously a good choice. For RSV + SEV, pp3 has the best performance in both cases. These provide a reference for the selection of pp in the real application context. The pp6 seems to have resulted in better output in most cases, compared to the pp5.

## 5. Conclusions

This study is seen as largely of a proof-of-concept nature. The possibility and feasibility of using a portable NIR spectrometer to detect the presence of two typical respiratory viruses RSV and SEV have been demonstrated via AI modelling along with data pre-processing, through (1) the accuracy (averaged for 10-runs) of 0.88, 0.94, and 0.93 for RSV, SEV, and RSV + SEV respectively using NN approach, of taking 3 sessions data in turn for training and the other one ‘unknown’ dataset for testing, which makes significant real prediction sense, and (2) the average accuracy of 0.94 (RSV), 0.97 (SEV), and 0.97 (RSV + SEV) for model validation and 0.90 (RSV), 0.93 (SEV), and 0.91 (RSV + SEV) for model test, with totally the 4 datasets, using 2 for model training, one for model validation, and the other one for model test. This demonstration of the possibility and feasibility provides the groundwork for a portable NIR spectrometer to be deployed for respiratory virus detection. The relatively weak quality of the instrument in detection has been demonstrated to be remediable by AI and machine learning approaches. At the current stage, an affirmative answer to the hypothesis presented in the Introduction section appears to stand.

This study confirms in a sense that a low-cost and portable spectrometer as used in this work is incapable of catching the NIR spectroscopic biomarkers of respiratory viruses with stable reproducibility, as likely limited by its intrinsic property. However, two issues need to be distinguished: to detect the spectroscopic biomarkers of the viruses and to detect the presence of the viruses. For the first one, it can be seen as a tentative assumption that the biomarker cannot be detected reproducibly by the low-cost instrument, which shall be tested in further research; and for the second one, it appears to be able to be realised independently of the former, i.e., the detection can be achieved by ‘bypassing’ the biomarker detection. In fact, finding out the biomarker or fingerprint is a Sufficient Condition to reach the goal of detection, rather than a Necessary Condition. There can be more than one way to reach the target; the AI approach and finding out visually the fingerprint are both sufficient conditions, but none of them is a necessary condition without which the target cannot be reached. The powerfulness of AI approaches is embodied here in that they can find and distinguish patterns through seemingly chaotic and visually unclear features. AI, as one of the Sufficient Conditions parallel to the fingerprint method, gives another way to get to the target without obtaining a visual feature illustration of the separation that a fingerprint would probably show. We would see that this is a contribution, a point of value of this study. On the other hand, the application of a lab-based benchtop High-Resolution NIR spectrometer may be able to find out the fingerprint and then manifest the possible reproducibility, proceeding to the detection in a different way in a more spectroscopy sense. A comparison of the outcome of a benchtop spectrometer with that of a portable one may also play a role in the validation of the effectiveness of the AI approaches. This can be a valuable practice and shall be considered in further investigation in this field as a more comprehensive study following the current pilot one.

The existence of the optically detectable difference, found in examination of the difference of virus-presence versus virus-absence (V1 vs. V0), that revealed the NIR characteristic properties of RSV and SEV, although with a small magnitude, exhibits a physical ground for realisation of the detection using the AI approaches, and the weak reproducibility, on the other hand, as seen to be an output of the simulation of the portable NIR spectrometer’s performance in a rough working condition, highlighted the significance of the capability of the AI approach in successful detection.

The VIP-Q algorithm presented in this study, i.e., the VIP scores along with the Quantile selection processing, has different functioning compared to the common usage of the VIP scores. This algorithm can apparently improve the performance of the AI models and is expected to work as a promising method in cognate studies. The truncation treatment can improve the output to some extent for detection and is of considerable significance as well. The advancement and utilisation of this specific variable selection algorithm enable this study to reach its goal and will be beneficial to further research. They may be seen as a special and useful form of pre-processing and pre-training.

Technically, it is notable that, (1) in the approaches used for the spectral data classification, NN performed the best and is recommendable to the similar study; (2) it is important to design/select a proper protocol of processing, including the method of pre-processing and the algorithm, to effectively classify the analytes; this study shows that pp3 and pp4 and pp6 are better choices for RSV, pp2 and pp3 are the best for SEV and RSV + SEV, respectively. (3) There was a ’reverse feature’, i.e., there are cases (Table 3 and Table 4) in which selecting wavelength variables corresponding to a small amount of higher VIP scores does not result in better models in the classification, but the practice of enclosing more wavelength points with VIP scores above a basic quantile line to exclude low VIP score points may yield a better subset for training and better models for classification with good accuracy–in NIR spectroscopy, incorporating as much spectral information as possible into the model may result in better models for a type of classification. This represents a difference from the functioning of conventional feature selection.

By accumulating experience in detecting viruses via the portable NIR spectrometer, this study provides a meaningful reference for further studies of SARS-CoV-2 detection with the convenient instruments of the category. The provision of the ground for the application is especially significant given that the SARS-CoV-2 and its various mutated viruses are spreading fast over the world, meeting the demand for a respiratory virus detection method with low cost, rapid speed, adequate accuracy, and suitability for being applied to a large population in both clinical and field settings.

Future work will also include more experiments to find possible better subinterval sets in the wavelength axis by adjusting the quantile value Q to obtain a more optimised sub-dataset for training. Ultimately, this study will provide the basis for detecting SARS-CoV-2 and other respiratory viruses via a portable NIR spectrometer underpinned by AI approaches.

Looking into the distance, we will evaluate the method on clinical virus samples (SARS-CoV-2 or/and its variants) and develop effective AI models for analysis that can be distributed through the internet for easy access. It is our vision that human saliva samples can be collected at locations where mass screening is needed, such as airports and train/bus stations. The human samples can be scanned with a portable NIR spectrometer, and the spectra can be analysed on site with results available in seconds. This will provide a fast, low cost, and easy-to-use respiratory virus screening method as an alternative to a thermometer gun.

## Figures and Tables

**Figure 1 sensors-24-00308-f001:**
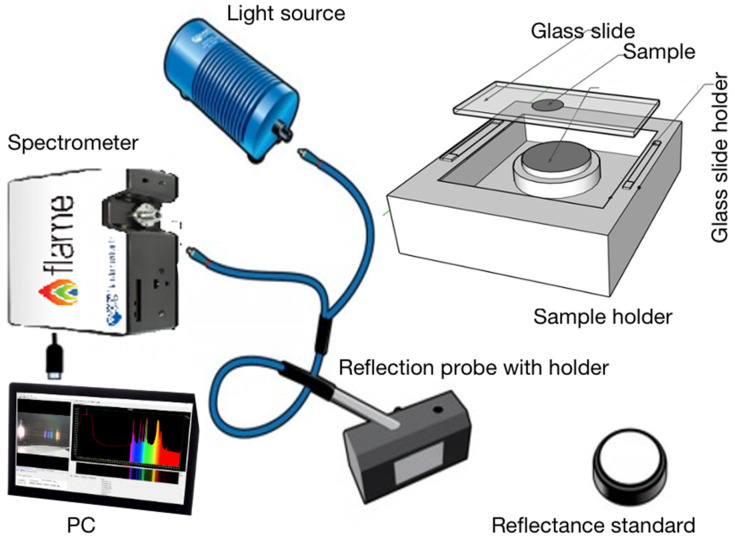
A schematic representation of the NIR spectrometer system used.

**Figure 2 sensors-24-00308-f002:**
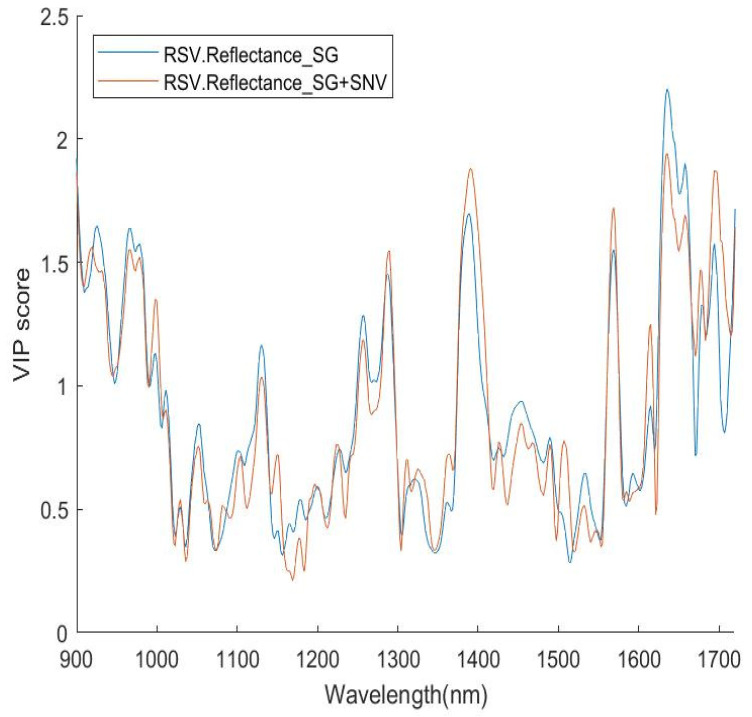
Examples of VIP score plots. Data were pre-processed by pp1 and pp2.

**Figure 3 sensors-24-00308-f003:**
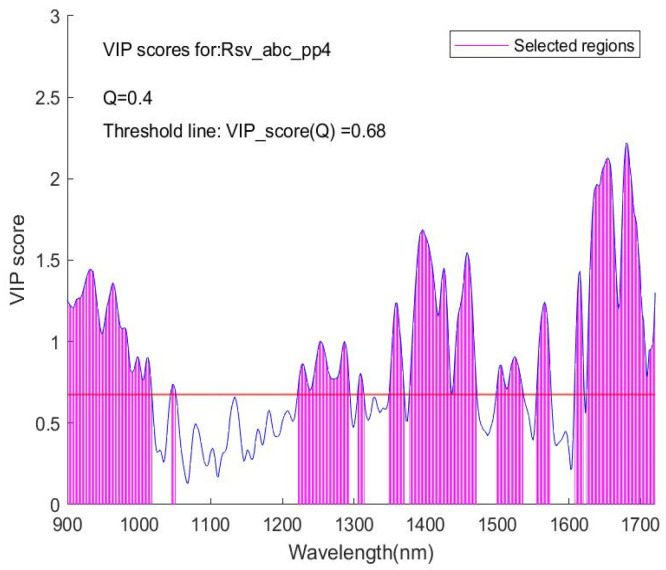
Illustration of the algorithm for feature selection via VIP scores.

**Figure 4 sensors-24-00308-f004:**
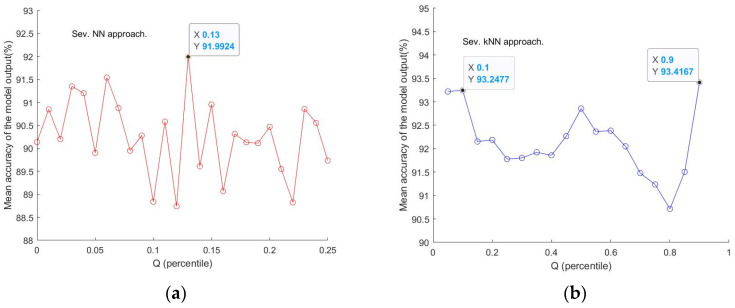
Illustration of the VIP-score-Q algorithm: to fix a Q that leads to an output with the maximum accuracy, (**a**) NN approach (SEV, V1 vs. V0) with Q = 0.13 selected. (**b**) kNN approach (SEV, V1 vs. V0) with Q = 0.1 or Q = 0.9 selected.

**Figure 5 sensors-24-00308-f005:**
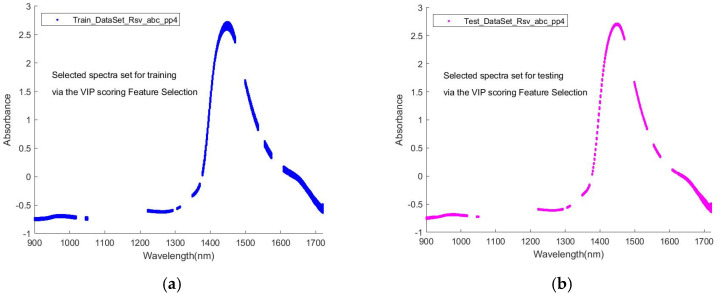
Illustration of selected sub-datasets for training and testing via VIP scoring and a Q value (**a**) The subset of a full set of spectra data, selected via treatment in Figure 4b for training and (**b**) the corresponding subset of the full set for testing.

**Figure 6 sensors-24-00308-f006:**
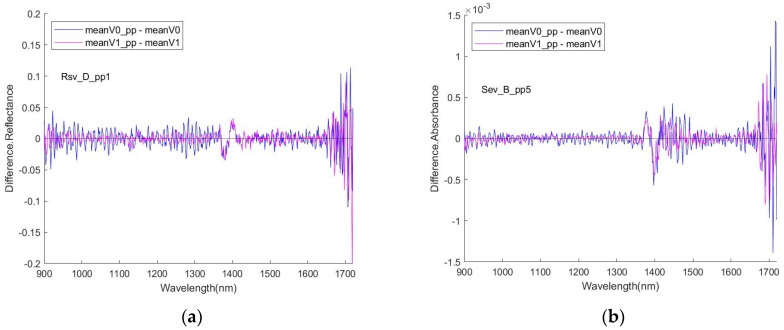
Removed components: difference between pre-processed and the original spectra, examples of (**a**) RSV and (**b**) SEV.

**Figure 7 sensors-24-00308-f007:**
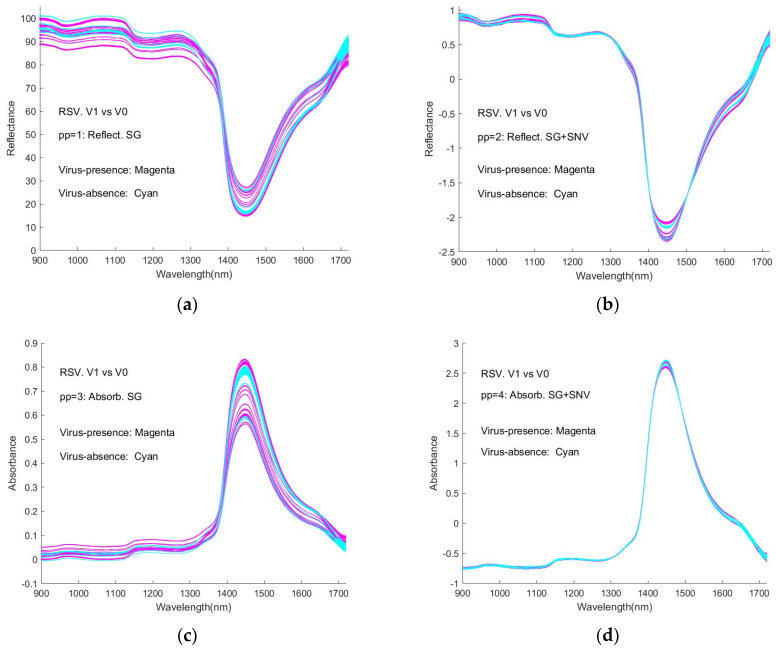
Plot of the RSV spectra after 6 pre-processing pipeline treatments (pp = 1 to pp = 6), V1 vs. V0. (**a**–**f**), corresponding to the pre-processing procedures pp1 to pp6 respectively, V1 (Magenta) vs. V0 (Cyan).

**Figure 8 sensors-24-00308-f008:**
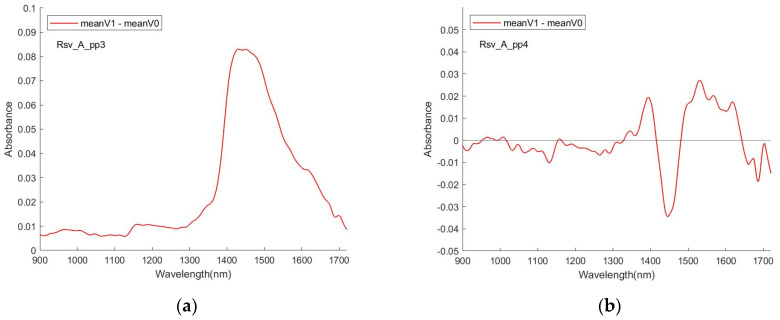
Difference of spectra: (meanV1–meanV0), pp3 to pp6. (**a**) Difference of spectra: (V1–V0), pp3. (**b**) Difference of spectra: (V1–V0), pp4. (**c**) Difference of spectra: (V1–V0), pp5. (**d**) Difference of spectra: (V1–V0), pp6.

**Figure 9 sensors-24-00308-f009:**
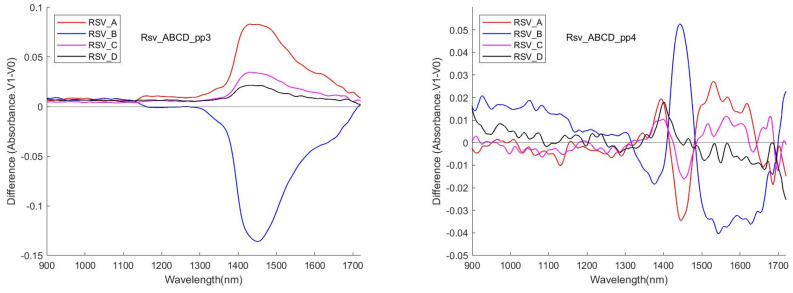
Examples of similarity/difference of (meanV1–meanV0) for the same pp for different sessions.

**Figure 10 sensors-24-00308-f010:**
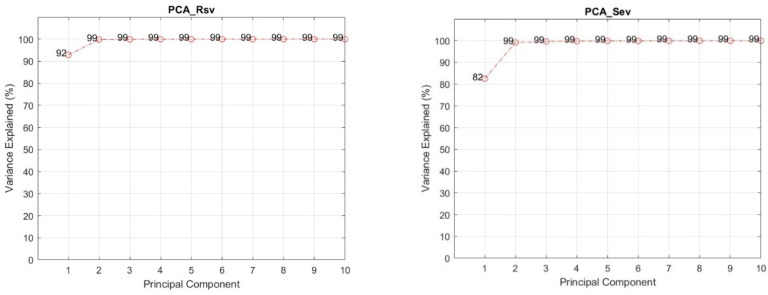
PCA for RSV and SEV, showing the Variance Explained (%) of the first to the tenth principal components.

**Figure 11 sensors-24-00308-f011:**
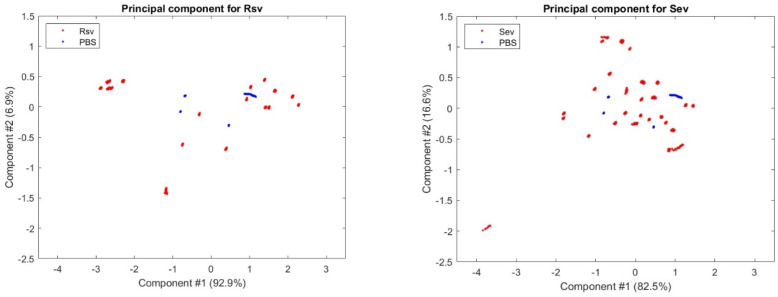
PCA scatter plots for the primary components of RSV and SEV.

**Figure 12 sensors-24-00308-f012:**
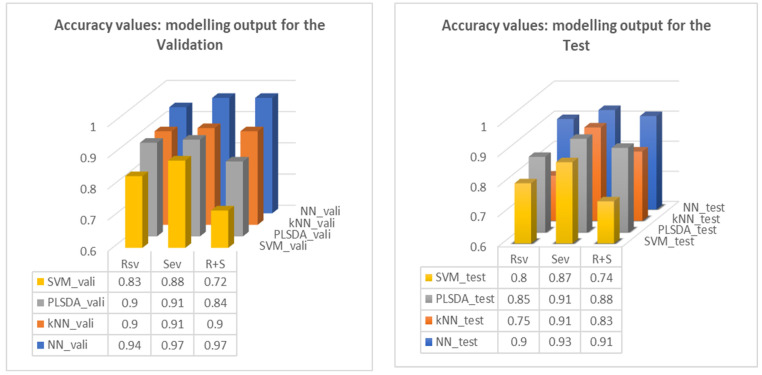
Accuracy values are used as the modelling output for the validation and test, respectively.

**Figure 13 sensors-24-00308-f013:**
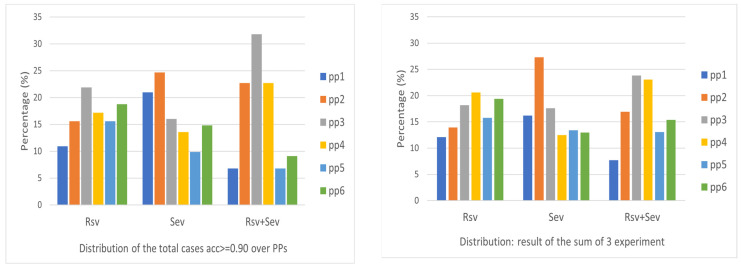
Distribution of good accuracy (≥0.9) over different pre-processing methods.

**Table 1 sensors-24-00308-t001:** Combination of the pre-processing (pp) methods used in this study.

Abbreviations	Combination of the Pre-Processing Methods
pp1	Reflectance, SG
pp2	Reflectance, SG + SNV
pp3	Absorbance, SG
pp4	Absorbance, SG + SNV
pp5	Absorbance, SG + MSC
pp6	Absorbance, SG + MSC + SNV

SG: Savitzky–Golay filter. SNV: Standard Normal Variate. MSC: Multiplicative Scatter Correction.

**Table 2 sensors-24-00308-t002:** Metrics for evaluating the model output.

Metrics	Calculation Formulae	Terms
Accuracy (ACC)	ACC = (TP + TN)/(TP + TN + FP + FN)	TP: number of true positives (virus-presence, v1) TN: number of true negatives (virus-absence, v0) FP: number of samples that are v0 but incorrectly classified as v1 (false positive) FN: number of samples that are v1 but incorrectly classified as v0 (false negatives)
Sensitivity (SENS)	SENS = TP/(TP + FN)	
Specificity (SPEC)	SPEC = TN/(TN + FP)	

**Table 3 sensors-24-00308-t003:** Accuracy: output of classification V1 vs. V0, based on the VIP scoring, (a) with SVM, (b) with PLSDA, (c) with kNN, and (d) with NN(MPL).

**(a)**									
	**RSV**			**SEV**			**RSV + SEV**		
	**(i)**	**(ii)**	**(iii)**	**(i)**	**(ii)**	**(iii)**	**(i)**	**(ii)**	**(iii)**
**Wavelength points set:** **1 to 512**	1.00 (Q = 0.50)	0.70 (Q = 0.50)	0.70 (Q = 0.50)	0.98 (Q = 0.90)	0.83 (Q = 0.90)	0.83 (Q = 0.90)	1.00 (Q = 0.60)	0.83 (Q = 0.90)	0.74 (Q = 0.80)
**Wavelength points set:** **1 to 450**	0.90 (Q = 0.80)	0.83 (Q = 0.80)	0.80 (Q = 0.80)	0.96 (Q = 0.70)	0.82 (Q = 0.80)	0.81 (Q = 0.80, 0.90)	1.00 (Q = 0.40)	0.85 (Q = 0.40)	0.85 (Q = 0.10)
**(b)**									
	**RSV**			**SEV**			**RSV + SEV**		
	**(i)**	**(ii)**	**(iii)**	**(i)**	**(ii)**	**(iii)**	**(i)**	**(ii)**	**(iii)**
**Wavelength points set:** **1 to 512**	1.00 (Q = 0.20)	0.87 (Q = 0.30, 0.90)	0.87 (Q = 0.30, 0.90)	1.00 (Q = 0.60, 0.90)	0.95 (Q = 0.90)	0.95 (Q = 0.90)	1.00 (Q = 0.10)	0.87 (Q = 0.10)	0.87 (Q = 0.10)
**Wavelength points set:** **1 to 450**	1.00 (Q = 0.25)	0.87 (Q = 0.45)	0.87 (Q = 0.45)	1.00 (Q = 0.75, 0.80)	0.90 (Q = 0.95)	0.90 (Q = 0.95)	1.00 (Q = 0.45, 0.50)	0.88 (Q = 0.45)	0.88 (Q = 0.45)
**(c)**									
	**RSV**			**SEV**			**RSV + SEV**		
	**(i)**	**(ii)**	**(iii)**	**(i)**	**(ii)**	**(iii)**	**(i)**	**(ii)**	**(iii)**
**Wavelength points set:** **1 to 512**	1.00 (Q = 0.05, …, 0.85)	0.86 (Q = 0.85)	0.80 (Q = 0.85)	1.00 (Q = 0.05, 0.10, 0.40, …, 0.90)	0.95 (Q = 0.05)	0.93 (Q = 0.90)	1.00 (Q = 0.02)	0.92 (Q = 0.01)	0.91 (Q = 0.04)
**Wavelength points set:** **1 to 450**	1.00 (Q = 0.90)	0.89 (Q = 0.90)	0.87 (Q = 0.90)	1.00 (Q = 0.80)	0.96 (Q = 0.80)	0.94 (Q = 0.80)	1.00 (Q = 0.05, 0.10)	0.93 (Q = 0.05, 0.10)	0.91 (Q = 0.05, 0.10)
**(d)**									
	**RSV**			**SEV**			**RSV + SEV**		
	**(i)**	**(ii)**	**(iii)**	**(i)**	**(ii)**	**(iii)**	**(i)**	**(ii)**	**(iii)**
**Wavelength points set:** **1 to 512**	1.00 (Q = 0.55, …, 0.65)	0.95 (Q = 0.57)	0.86 (Q = 0.61)	1.00 (Q = 0.01, …, 0.25)	0.97 (Q = 0.23)	0.92 (Q = 0.13)	1.00 (Q = 0.75, …, 0.85)	0.95 (Q = 0.78)	0.91 (Q = 0.82)
**Wavelength points set:** **1 to 450**	1.00 (Q = 0.85)	0.95 (Q = 0.75, 0.85)	0.88 (Q = 0.85)	1.00 (Q = 0.80)	0.99 (Q = 0.80)	0.94 (Q = 0.80, 0.90)	1.00 (Q = 0.90)	0.95 (Q = 0.90)	0.93 (Q = 0.90)

(i). Maximum of A (i, j) in 10 successive runs (i = 1 to 6, j = 1 to 4). (ii). Maximum of Mean_AccPPmax in 10 successive runs, corresponding to a Q. (iii). Average of Mean_AccPPmax in 10 successive runs.

**Table 4 sensors-24-00308-t004:** Modelling output with different combinations of datasets for training, validation, and testing. (a) with SVM and PLSDA. (b) with kNN and NN.

**(a)**								
**Data Sets for the Model Training**	Data Set for the Model Validation	Data Set for the Model Test	SVM Modelling Output: Accuracy//Q			PLSDA Modelling Output: Accuracy//Q		
			RSV	SEV	RSV + SEV	RSV	SEV	RSV + SEV
**A + B**	C		0.80//0.6	0.88//0.1	0.85//0.2	0.83//0.6	0.88//0.1	0.85//0.8
**A + B**		D	0.99//0.6	0.89//0.1	0.87//0.2	0.98//0.6	0.89//0.1	0.96//0.8

**A + C**	B		0.80//0.1	0.95//0.2	0.78//0.1	0.95//0.8	0.99//0.5	0,89//0.2
**A + C**		D	0.80//0.1	0.90//0.2	0.90//0.1	0.80//0.8	0.90//0.5	0.86//0.2

**A + D**	B		1.00//0.9	1.00//0.1	0.93//0.8	1.00//0.3	1.00//0.1	0.86//0.1
**A + D**		C	0.80//0.9	0.88//0.1	0.85//0.8	0.80//0.3	0.88//0.1	0.85//0.1

**B + C**	A		0.80//0.9	0.99//0.9	0.73//0.7	0.80//0.6	1.00//0.1	0.73//0.7
**B + C**		D	0.80//0.9	0.89//0.9	0.14//0.7	0.94//0.6	1.00//0.1	0.99//0.7

**B + D**	C		0.80//0.9	0.85//0.9	0.30//0.1	0.80//0.5	1.00//0.2	0.73//0.4
**B + D**		A	0.60//0.9	0.75//0.9	0.85//0.1	0.80//0.5	0.88//0.2	0.77//0.4

**C + D**	A		0.80//0.1	0.58//0.1	0.73//0.1	1.00//0.4	0.58//0.1	0.95//0.1
**C + D**		B	0.80//0.1	0.88//0.1	0.85//0.1	0.80//0.4	0.93//0.1	0.85//0.1

**Average**	Validation		0.83	0.88	0.72	0.90	0.91	0.84
**Average**		Test	0.80	0.87	0.74	0.85	0.91	0.88
**(b)**								
**Data Set for the Model Training**	Data Set for the Model Validation	Data Set for the Model Test	kNN Modelling Output: Accuracy//Q			NN Modelling Output: Accuracy//Q		
			RSV	SEV	RSV + SEV	RSV	SEV	RSV + SEV
**A + B**	C		0.80//0.1	0.88//0.1	0.85//0.1	0.80//0.5	0.88//0.1	0.85//0.5
**A + B**		D	0.80//0.1	0.89//0.1	0.86//0.1	1.00//0.5	0.89//0.1	0.86//0.5

**A + C**	B		1.00//0.1	1.00//0.2	1.00//0.2	1.00//0.6	1.00//0.1	1.00//0.5
**A + C**		D	0.80//0.1	0.96//0.2	0.86//0.2	0.97//0.6	0.99//0.1	0.94//0.5

**A + D**	B		1.00//0.1	1.00//0.2	1.00//0.1	1.00//0.1	1.00//0.8	1.00//0.1
**A + D**		C	0.80//0.1	0.88//0.2	0.85//0.1	0.80//0.1	0.91//0.8	0.85//0.1

**B + C**	A		0.80//0.6	1.00//0.1	1.00//0.1	1.00//0.7	1.00//0.4	0.96//0.6
**B + C**		D	0.73//0.6	0.95//0.1	0.86//0.1	0.83//0.7	0.99//0.4	0.96//0.6

**B + D**	C		0.80//0.8	1.00//0.6	0.73//0.7	1.00//0.9	1.00//0.9	0.99//0.7
**B + D**		A	0.60//0.8	0.88//0.6	0.69//0.7	0.80//0.9	0.81//0.9	0.83//0.7

**C + D**	A		1.00//0.7	0.58//0.2	0.83//0.7	0.81//0.7	0.92//0.4	1.00//0.8
**C + D**		B	0.79//0.7	0.88//0.2	0.85//0.7	0.98//0.7	0.97//0.4	1.00//0.8

**Average**	Validation		0.90	0.91	0.90	0.94	0.97	0.97
**Average**		Test	0.75	0.91	0.83	0.90	0.93	0.91

Note: A, B, C, and D represent the datasets collected on 15 October 2020, 29 October 2020, 20 November 2020, and 10 December 2020, respectively.

## Data Availability

Data are contained within the article.

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
