# Peer review of "Detecting Respiratory Viruses Using a Portable NIR Spectrometer—A Preliminary Exploration with a Data Driven Approach"

_sensors, 2024, doi:10.3390/s24010308_

Round 1
Reviewer 1 Report (New Reviewer)
Comments and Suggestions for Authors
This article describes an interesting and intriguing concept: Is it possible to detect virus in solution with rapid and low-cost near-infrared spectroscopy, so that the method can be used for effective screening and diagnosis?
The study is heavily data driven – as stated in the title – and that is the major limitation of this work. The most interesting and important part of a study (or concept) like this would be to document the underlying mechanism for the sensor system: What do we actually measure? What chemical bonds is it possible to pick up from virus with NIR spectroscopy. The authors have not discussed this at all. There are no hypothesis or discussions at all regarding the realism in being able to register signals from viruses with NIRS – with expensive lab-instrument or low-cost. They find no support for this in the literature, nothing described in the introduction or discussed in the discussion.
The authors write that common limitations with NIRS are low resolution and high noise. This is not correct. NIR systems can have high or low resolution, but the specificity of the method is low. Some sort of chemometrics or multivariate modelling is normally required. NIR systems do normally have very high signal-to-noise ratio (very little noise), this is one of the big advantages with NIR. A poorly used hand-held system can of course produce noisy spectra. But: a severe limitation with NIRS that the authors do not mention, is that the method is not very sensitive. Limit of detection in complex systems is typically 0.5%. That is why you do not find many articles trying NIRS for virus detection – it is most likely not a feasible method. While SERS, as mentioned in the intro, is much more sensitive and might work.
I am not an expert in viruses. But let’s say a maximum load of virus would equal about 0.0001g/ml = 0.01%. This is far below what you can expect to detect with NIRS. Especially when the virus is diluted in water. Water absorbs heavily in the NIR and it is well known that is difficult to detect small concentrations of substances in water. Raman or fluorescence is much better, since these methods are not sensitive to water. All your figures show absorption spectra of water, and there are no indications that you have seen anything else. In the description of your samples, you should indicate the approximate amount of gram virus per ml. Then it is easier to understand the sensitivity requirements.
To be able to publish this, you need to present at least some spectral signatures from your samples that are not only water. If your classification results are real and soundly based on chemistry, and not overfitted, then you should absolutely be able to detect the spectral signatures that enables this result. Just a simple PCA of your spectra where you can show scores that to some degree correspond with your mixture design, in combination with loadings that can reveal the responsible spectral features could be enough. In this article you show only difference spectra between mean spectra of the different groups. These are not interesting, they would be based on just random variations.
All the different classification methods and pre-processing methods you are using here are of little interest. It is the main spectral information that is important at this stage. There is no need to optimize the machine learning algorithms as long as you have no clue about the fundamental spectroscopy of your sensor system. I would suggest that you focus on understanding the spectroscopy. Then it is sufficient to use just one classification method and one type of pre-processing to demonstrate that this actually works.
As long as you are doing classification and not regression, you can use reflectance just as well as absorption. They contain the same information. Absorption is used in regression to linearize the spectral data with regard to chemical reference values.
As the paper is written now, the data analysis is too massive and appears as a non-transparent wall. It is difficult to judge whether the validation is done properly. You claim to get very promising results, but they are difficult to trust since
1. There is no underlying rationale for why it should work.
2. The data analysis and especially the validation regime is too vaguely described.
To me it seems like datasets A, B, C and D are completely similar except for that they were made on different days. Are they then independent samples?
Explain better how the protocol in lines 104-113 is connected to measurements described in 114-117. This is not clear. A figure can be used to show an overview of the samples, number of samples/measurements (50 measurements per sample??), and how you ensure that test samples are completely independent from the calibration sets.
Author Response
Response to the Reviewers
Journal: Sensors (ISSN 1424-8220)
Manuscript ID: sensors-2610251
Type: Article
Title: Detecting respiratory viruses using a portable NIR spectrometer – A preliminary exploration with data driven approach
Authors: Jian-Dong Huang *, Hui Wang *, Ultan Power, James A. McLaughlin, Christopher Nugent, Enayetur Rahman, Judit Barabas, Paul Maguire
Section: Biosensors
Response to the Reviewer 1
Reviewers’ comments on the manuscript, and the authors’ answers:
|
Reviewer’s comments (Reviewer1) |
Authors’ response Notes: in the text below, ‘the old version’ means the version submitted on 30/08/2023, and ‘the new version’ means the revised version re-submitted this time. |
|
Paragraph 1 This article describes an interesting and intriguing concept: Is it possible to detect virus in solution with rapid and low-cost near-infrared spectroscopy, so that the method can be used for effective screening and diagnosis?
|
Response1: Thank you for your positive comments. We have addressed your concerns by carrying out revising the relevant sections.
General statements: The logic of this study is, 1) To cope with the large scale respiratory infectious disease like the pandemic of COVID-19 that threaten human health, methodology for detecting the viruses is essential. 2) The lab-based methods, including RT PCR, ELISA, CGIA, and SERS etc., have their shortcomings in terms of the cost, potability, ease of use, especially the rapid mass deployment in both clinical and field settings. 3) NIR methods include lab-based and portable ones, the former also has similar shortcomings as stated above; thus, this study is exclusively focused on the latter, for making use of the latter’s merits and avoiding the former’s shortcomings. 4) As the spectra from a portable NIR spectrometer demonstrated interweaved curves of virus-absent and virus-present samples, we encountered huge obstacle in distinguishing between the two, leading to a possibility that the trial might end up with failure. 5) AI techniques were then deployed to try to solve the problem. But after the attempt of using several algorithms to model the data, the output was not optimistic --- the accuracies were only around 0.7 ~ 0.8. This forms another huge obstacle of the study. 6) To overcome the hinder, an advanced VIP-Q algorithm combined with the spectra truncation method were applied, with promising outcome. We see this is a significant contribution of this study.
Thank you for clearly stating the hypothesis of this study which was implicitly expressed in the Introduction and Conclusion sessions: ‘’Is it possible to detect virus in solution with rapid and low-cost near-infrared spectroscopy, so that the method can be used for effective screening and diagnosis?’’ Our work with 4) to 6) of the above has achieved such a vision that a positive answer to the hypothesis has not been falsified so far and a way of solving the issue along this direction is hopeful. Consequently, this work, with its possible application prospects, is worthy of being reported to the public, to appeal and encourage the peers and the research/industry community to try further investigations, and to benefit the society if the later deterministic outcome would be promising. This is a limited study in terms of its scope and time span, we just wish to present our preliminary results at this stage rather than to present a ‘perfect’ investigation result. An understanding to our proof-of concept-study shall be greatly appreciated.
The above thinking is reflected in the Introduction and other relevant section(s) of the revised version of the manuscript. |
|
Paragraph 2
The study is heavily data driven – as stated in the title – and that is the major limitation of this work. The most interesting and important part of a study (or concept) like this would be to document the underlying mechanism for the sensor system: What do we actually measure? What chemical bonds is it possible to pick up from virus with NIR spectroscopy. The authors have not discussed this at all. There are no hypothesis or discussions at all regarding the realism in being able to register signals from viruses with NIRS – with expensive lab-instrument or low-cost. They find no support for this in the literature, nothing described in the introduction or discussed in the discussion. |
Response2:
Thanks for your comments. We have addressed the concern in the revised version.
From a view of AI, the spectra are only a medium, an information carrier for virus-presence or virus-absence. The Figure 8 and Figure 9 in both of the old version and the new version have shown that there IS some component regarding the existence of the virus(es), represented in the spectra. Thus, as long as we can catch and distinguish the (two types of) information via the sophisticated AI algorithm, we then can reach our goal and achieve our aim, through the treatment of the variable selection and the NN modelling. Looking into the chemical side shall be one of the good ways to reach our detection goal but is not the ONLY way to reach it. Investigation via the chemical/structural details is not the focus of this study but can be a good theme in further studies within this field. As stated in the ‘General statements’, there is really a hypothesis of this study in a general-level, and the positive answer to the hypothesis has not been falsified, leading to the value of reporting this study to the peers. For the examination in a detailed level such as the mechanism in spectroscopy and chemistry, because our goal is in such a level to test the general hypothesis, it appears adequate to mainly focus on grasping the discriminatory information hidden in the spectra signals without missing it (via the AI means) and leave the exploration on spectra/chemical analysis-level to the future.
Please also see the ‘Response8 for the Paragraph 8’ below for the rationale of using the AI approaches.
As an exploratory study, it would be not surprising that rare or no corresponding literature records can be found in the literature. This can be a reason that this work is seen as a pilot one and worthy of being highlighted.
This is an urgent project requesting a fast output within a relative short time span. Therefore, a very limited objective was set as to explore and demonstrate the possibility and feasibility of using the portable instrument to detect the presence of the viruses through processing/analysing the NIR spectra samples with AI approaches. The designed model establishing/testing process as a significant component of the study have demonstrated good quality of the obtained models (Tables 3 and 4). |
|
Paragraph 3
The authors write that common limitations with NIRS are low resolution and high noise. This is not correct. NIR systems can have high or low resolution, but the specificity of the method is low. Some sort of chemometrics or multivariate modelling is normally required. NIR systems do normally have very high signal-to-noise ratio (very little noise), this is one of the big advantages with NIR. A poorly used hand-held system can of course produce noisy spectra. But: a severe limitation with NIRS that the authors do not mention, is that the method is not very sensitive. Limit of detection in complex systems is typically 0.5%. That is why you do not find many articles trying NIRS for virus detection – it is most likely not a feasible method. While SERS, as mentioned in the intro, is much more sensitive and might work.
|
Response3:
Thank you for pointing out this. We have made corrections accordingly in the text (Introduction section of the new version). We exclusively focused on the portable NIR spectrometers without including the lab-based ones, but it was ignored to emphasise that this was a portable one in the relevant text. We are sorry for the carelessness and grateful to the reviewer for the mentioning and correction.
The models established by the ML algorithms are multivariable (512 dimension) models. Sorry we need to mention this in the text. Thank you for pointing out that a severe limitation with NIRs that the authors do not mention, is that the method is not very sensitive. This has been revised in the text in the new version (Introduction section, the new version).
This work, so far, has demonstrated that it is possible to detect the viruses through the NIRs supported with the AI approaches, which then appears helpful to change the vision that the portable NIR spectrometer would not be a feasible method for the viruses’ detection.
|
|
Paragraph 4
I am not an expert in viruses. But let’s say a maximum load of virus would equal about 0.0001g/ml = 0.01%. This is far below what you can expect to detect with NIRS. Especially when the virus is diluted in water. Water absorbs heavily in the NIR and it is well known that is difficult to detect small concentrations of substances in water. Raman or fluorescence is much better, since these methods are not sensitive to water. All your figures show absorption spectra of water, and there are no indications that you have seen anything else. In the description of your samples, you should indicate the approximate amount of gram virus per ml. Then it is easier to understand the sensitivity requirements.
|
Response4:
The load of the viruses was 0.4 –1.0 g/ml: in the line 115, page 3 in the old version, and in the line 164, page 4 in the new version.
Water absorbance was discussed in line 394—408, page 12; line 616-620, page 19, of the old version. The correspondence in the new version are: line 454-468, page 12-13; line694-697, page 20, of the new version.
There are Figures (Fig.8 and Fig.9) in the old version and the new version, showing the difference between the spectra of virus-present and virus-absent samples, i.e., the curves after removing the signal of solution (PBS) from the signal of solution addition (virus + PBS). The curves are not flat as a horizontal line, clearly indicating that there is something detected by the spectra of the NIR spectrometer, representing that the significant information has been caught, which forms a basis of the further AI distinguishing between the two.
The approximate amount of the gram virus is indicated in the line 115 in page 3, in the old version, and line 164, page 4, in the new version.
Please also see the section 4.1, line 522-568, page 17 to 18 in the old version, and line 598-646, page 19-20 in the new version.
|
|
Paragraph 5 To be able to publish this, you need to present at least some spectral signatures from your samples that are not only water. If your classification results are real and soundly based on chemistry, and not overfitted, then you should absolutely be able to detect the spectral signatures that enables this result. Just a simple PCA of your spectra where you can show scores that to some degree correspond with your mixture design, in combination with loadings that can reveal the responsible spectral features could be enough. In this article you show only difference spectra between mean spectra of the different groups. These are not interesting, they would be based on just random variations. |
Response5:
Fig.8 and Fig.9 on pages 12 and 13 in the old version submitted on 03/08/2023 (page 13 and 14 in the new version) respectively represent spectral signatures from the samples that are not water.
The validation results, especially for the Neural Network algorithm (all above 0.9 for the accuracy), indicating the good quality of the classification.
Following your suggestion, PCA graphs have been plotted and added (Fig. 10 and Fig. 11, page 15 in the new version), showing the first and the second primary component for Rsv and Sev respectively, which are helpful to provide ground of the AI processing from a different angle. Many thanks.
The mean spectra were used to demonstrate the difference between virus +solution (PBS) and the solution (PBS) only, not different virus groups. Because of the large number of spectra data, we just used the mean values to represent the general situations.
We understand that a mean value of large amount quantities is commonly used to remove error in a statistical sense and is used to avoid possible biases.
|
|
Paragraph 6 All the different classification methods and pre-processing methods you are using here are of little interest. It is the main spectral information that is important at this stage. There is no need to optimize the machine learning algorithms as long as you have no clue about the fundamental spectroscopy of your sensor system. I would suggest that you focus on understanding the spectroscopy. Then it is sufficient to use just one classification method and one type of pre-processing to demonstrate that this actually works.
|
Response6: Thanks for the comments. In our view, the spectra of portable spectrometer of NIRs just play a role of a virus information carrier. On the one hand, the spectra tell us there is really some signature in existence, demonstrated in the Fig.8 and Fig.9 through the difference between virus + PBS and PBS, which forms a firm basis for the AI approaches to work on and to find the answer of the detection via classification; on the other hand, due to the relatively poor quality of the instrument, reflected in the poor reproducibility of a stable feature of the signature, then, merely using the spectroscopy to realise the detection has encountered a huge and severe hinder. As such, the former provides a possibility, the latter leads to a necessity, to use AI approaches to overcome the difficulty or bypass the intrinsic defect of the instrument to find the needed answer of the detection. This is the logic of the study --- bypassing the reproducibility issue with relation to the spectroscopy of the relatively poor-quality instrument and find the answer we need. Therefore, our study was mainly focused of the AI side rather than the spectroscopy side with the obstacle. This is stated in the relevant part of the revised version of the manuscript. Given that our approach is AI based, it seems better to provide the peers with details of the performance of different algorithms, and the results under different pre-processing of the spectra, for the algorithm selection and model validation/test. That is why we present the rich results for the readers to see which one is relatively better.
|
|
Paragraph 7 As long as you are doing classification and not regression, you can use reflectance just as well as absorption. They contain the same information. Absorption is used in regression to linearize the spectral data with regard to chemical reference values. |
Response7: Yes, we do agree with the important comments about information contained in the spectra. On the other hand, different pre-processing methods (associated with reflectance or absorbance) leads to different levels of the modelling accuracy (Fig 12, page 21 in the old version and Fig.13, page 22 in the new version), and we need to compare and report the performance of the methods we tried, to inform the peers of the effects of the different methods, for them to be benefited more from this study.
|
|
Paragraph 8 As the paper is written now, the data analysis is too massive and appears as a non-transparent wall. It is difficult to judge whether the validation is done properly. You claim to get very promising results, but they are difficult to trust since 1. There is no underlying rationale for why it should work. 2. The data analysis and especially the validation regime is too vaguely described. |
Response8: For the rationale of the approach, two points can be raised here: 1) There are great number of literatures that demonstrate the effectiveness and reliability of neural networks to approximate real word objects (or curves), such as the Theory of the Backpropagation Neural Network, by R. Hecht-Nielsen, that indicated “any L2 function can be implemented to any desired degree of accuracy with a three-layer backpropagation neural network”. This means, no matter how complex a curve is like, there is a way to approximate it through the AI approaches. The earlier fundamental works like this formed the ground of our belief and confidence that the AI approaches shall obtain trustable outcomes, as evidenced by our results. 2) The figure Fig.8 and Fig.9 have shown that there is some meaningful information detected in the spectra, for the AI approach to work on and to find the relations behand, with the confidence from the above mentioned. 3) Our data analysis follows a de-facto protocol in machine learning. Data is split into training and validation/testing sets by sessions. Feature selection is done by VIP-Q. Modelling is done using different algorithms (SVM, PLSDA, kNN, and NN). Evaluation is done using accuracy metric. Validation is done by visualisation of yes/no spectral difference, as well as visualisation of data by PCA components (thanks for the suggestion). Since NIR is reflectance spectroscopy, there is no virus spectral lines as in emission spectroscopy. The above is reflected in the Introduction section and later relevant sections, of the revised version. We hope the above can be of help to clarify the issue of rationale of the study. Yes, it seems difficult to balance how detail the describing of the AI methodology is given in the text. From a view of AI approach, when undertaking the modelling with a conventional way, model establishing –> validation –> testing, is a common practice that needs only a brief mention. The steps were described there but, to follow the suggestion, some more explanation has been added and the relevant text re-organised/re-written.
|
|
Paragraph 9 To me it seems like datasets A, B, C and D are completely similar except for that they were made on different days. Are they then independent samples? |
Response9: It is understandable that the independence of the samples would be likely a concern. One of the co-authors who works on virology for decades, gave statements as below (mainly in line 548-568, page 18 of the old version, and line 625-646, page 19-20 in the new version): “With due respect to the reviewer, when establishing new techniques or experiments with viruses, it is best practice in virology to prepare a single stock of virus that is aliquoted and stored for repeated use during the protocol validation. Virus stocks are crude preparations derived from infected cells and each stock preparation will differ to some degree in terms of the quantifying of infectious and non-infectious virions. As such, in order to validate the ability of the technology to repeatedly detect individual virus species and distinguish between different virus species, it is imperative to use the same stocks of virus repeatedly. A single stock of each virus species that is aliquoted and frozen allows for several independent experiments to be undertaken with the exact same material, which is essential to clearly demonstrate the capacity of the technology to repeatedly distinguish between the presence and absence of viruses under investigation. Our study demonstrated this ability. “ Within the analyte, the object we measured each time is one virus that has its structure and optical characteristics, which is fixed or firmly existed. Measurement of more samples will not change these, but only involve more environmental factors such as the temperature, the weather conditions, the surrounding vibration, and the possible slightly varied cultivation conditions of the virus(es) in the lab, and with difference ‘to some degree in terms of the quantifying of infectious and non-infectious virions.’ Consideration of these into the sample preparation would lead the models being trained under more ‘noises’ and would commonly lead to better representative output, but does not change what the fundamental properties of the target we want to detect as a main component. However, further to this proof-of-concept study, we wish to make deeper exploration under consideration of more rigorous data collection environment and conditions in the future. |
|
Paragraph 10 Explain better how the protocol in lines 104-113 is connected to measurements described in 114-117. This is not clear. A figure can be used to show an overview of the samples, number of samples/measurements (50 measurements per sample??), and how you ensure that test samples are completely independent from the calibration sets. |
Response10: Thanks for the important comments. The lines 104-117, page 3 of the old version, and149-166, page 3-4 of the new version, were written by experienced virologists after the implementation of the experiments, and it is believed that the description is clear and adequate to the peers. We wished to give some more content about this but considering that the paper has already a large size, may we wish to avoid more space-occupying content, and wish the reviewer would understand this, thank you.
Yes, there were 50 spectra obtained for each scan (line 91, page 2; line 560-564, page 18; line 607-608, page 19, in the old version; and line 140, page 3; line 637-642, page 19; line684-686, page 20, in the new version).
For the concern about the independency, please see the above “Response9”. As this is an urgent project and the study is a pilot one, the emphasis was mainly on the successful realisation of the detection, and the questions the reviewers raised might be temporarily put aside, which could be a topic of further study. We wish to report the results as an important progress in this research field in a timely manner and encourage further studies. |

Reviewer 2 Report (New Reviewer)
Comments and Suggestions for Authors
The authors employed the portable NIR spectrometer to study the two common respiratory viruses, the Respiratory syncytial virus (RSV) and the Sendai virus (SEV). By accumulating experience in detecting the viruses via the portable NIR spectrometer, this study provides a meaningful reference for further studies of SARS-cov-2 detection with the convenient instruments of the category. The results indicated that the NIR spectroscopy technique coupled with machine learning to detect respiratory viruses exhibited the good accuracy, and the approach could be a viable solution for population screening. I recommend it is publishable after major revision.
The following issues should be reconsidered.
1. The authors should emphasize the advantage of the NIR technique in the introduction part.
2. Why the authors employed the portable NIR spectrometer coupled with machine learning? This should be introduced in the manuscript.
3. Figures 7-1 and 7-2 are the Reflectance spectra, and Figures 7-3, 7-4, 7-5, and 7-6 are the Absorbance spectra, these spectra should be uniformed.
4. The figure captions should be explained clearly. The figures caption of Figure 7-1, 7-2..., Figure 8-1, 8-2..., should moved to the caption of Figure 7 and 8.
5. The Appendix part should add the figure caption in each group of figure.
Author Response
Response to the Reviewers
Journal: Sensors (ISSN 1424-8220)
Manuscript ID: sensors-2610251
Type: Article
Title: Detecting respiratory viruses using a portable NIR spectrometer – A preliminary exploration with data driven approach
Authors: Jian-Dong Huang *, Hui Wang *, Ultan Power, James A. McLaughlin, Christopher Nugent, Enayetur Rahman, Judit Barabas, Paul Maguire
Section: Biosensors
Response to the Reviewer 2
Reviewers’ comments on the manuscript, and the authors’ answers:
|
Reviewers’ comments (Reviewer 2) |
Authors’ response: |
|
The authors employed the portable NIR spectrometer to study the two common respiratory viruses, the Respiratory syncytial virus (RSV) and the Sendai virus (SEV). By accumulating experience in detecting the viruses via the portable NIR spectrometer, this study provides a meaningful reference for further studies of SARS-cov-2 detection with the convenient instruments of the category. The results indicated that the NIR spectroscopy technique coupled with machine learning to detect respiratory viruses exhibited the good accuracy, and the approach could be a viable solution for population screening. I recommend it is publishable after major revision.
|
Response:
Thank you for your positive comments.
We have addressed your concerns by carrying out revising and editing in the relevant sections |
|
The following issues should be reconsidered. 1. The authors should emphasize the advantage of the NIR technique in the introduction part. 2. Why the authors employed the portable NIR spectrometer coupled with machine learning? This should be introduced in the manuscript. 3. Figures 7-1 and 7-2 are the Reflectance spectra, and Figures 7-3, 7-4, 7-5, and 7-6 are the Absorbance spectra, these spectra should be uniformed. 4. The figure captions should be explained clearly. The figures caption of Figure 7-1, 7-2..., Figure 8-1, 8-2..., should moved to the caption of Figure 7 and 8. 5. The Appendix part should add the figure caption in each group of figure.
|
Response:
1. Thank you for your suggestion, the relevant content has been added into the Introduction section (line 66 to 123, page 2-3, in the newly submitted version). 2. This has been addressed in the Abstract, Introduction and other relevant section accordingly. 3. As part of planed investigation, we used 6 different spectra pre-processing methods, from pp1 to pp6 (Table 1 in page 4 of the old version and Table 1 in page 5 in the new version) --- the pre-processing of pp1 and pp2 worked on Reflectance spectra and the pp3 to pp6 worked on Absorbance spectra, we then have to report all of the results to the peers / readers, please understand this. 4. The suggestions have been followed, many thanks. 5. The suggestions have been followed, many thanks. |

Round 2
Reviewer 2 Report (New Reviewer)
Comments and Suggestions for Authors
The revised manuscript is recommened acceptable.
This manuscript is a resubmission of an earlier submission. The following is a list of the peer review reports and author responses from that submission.
Round 1
Reviewer 1 Report
Comments and Suggestions for Authors
Near infrared spectroscopy is emerging to be a powerful technique to solve biomedical problems. though the signals are weak compared to mid infrared, but when coupled with powerful machine learning algorithms It has the capacity to become a screening method for serums, blood and urine. The present manuscript could be a great example of such application. However, there are few things need to be addressed as given below.
1) There is a small typo in the line 168-instead of scores it says sores
2) What is the typical minimal amount of virus detected by conventional lab instrument?
3) How does NIR perform in that range?
4) I have a question about the methodology: Were all different virus dilutions labeled as ‘with virus’ and the buffer samples as ‘without virus’? Was it done for a specific reason?
5) Instead of taking three complete sets of samples to make a model and predicting with one set, what happens if you use 2/3rd of the data from each set and leave the 1/3rd of data from each set to predict, I think your PLSDA may perform better.
6) I am a big advocate of spectroscopic solutions to the biomedical problems and I understand this method will be very helpful as a prescreening method. However, could you please add a paragraph how you envision it to be used step by step in clinic or on the field.
Thank you.
Author Response
Thank you for your valuable comments. Amendments have been made in the revised version of the manuscript, to address your concern. Please see the attached file. Thank you.

Reviewer 2 Report
Comments and Suggestions for Authors
The paper aims at showing the potential of NIR spectroscopy and machine learning for the detection of respiratory viruses. However, the experimental design is rather poor and very far from the conditions which could be useful in routine analysis. Moreover, prepared samples are not independent on one another and therefore the results are completely overfitted and overoptimistic. Altogether, I think that the manuscript could be considered as suitable for publication in its present form and a very extensive revision is needed.
Below, I am detailing the main problematic issues:
1) End of page 3: I think that a PBS solution at relatively high concentrations (for a virus) is not the best system to simulate practical detection of viruses for routine real world applications.
2) End of page 3: Were the stock solution freshly prepared before each measurement session or was the same solution used across all the measurements?
3) End of page 3: "comprised a total of 2370 spectra containing RSV, SeV viruses in a PBS medium, across all dilutions, along 115 with samples containing PBS medium only". Considering what reported earlier, were the 2370 spectra recorded on as many independently prepared mixtures, or there were also replicated measurements of the same samples?
4) Section 2.3: "converted to absorbance using the Kubelka-Munk transformation (A = log10(1/ R))". This is not the Kubelka-Munk transformation, but simply a reflectance to pseudo-absorbance transformation.
5) Section 2.3 item 1. The authors speak of Savitsky-Golay differentiation, but in the remainder of the study they only use smoothing without derivation.
6) Section 2.3 item 2. "or linearly varying baseline off set, due to light-scattering". Light-scattering results in a multiplicative effect which is different than a linearly varying baseline. The latter can be simply corrected by second derivative or detrending.
7) Page 6, line 4: "VIPs identify the most relevant variables related to a dependent Y in an independent array X." VIP are defined only for PLS-DA models, so they can only be calculated based on PLS-DA. This should be specified (and maybe it could also be good to specify why interpretation/variable selection for the other ML technique wasn't carried out through technique-specific approaches)
8) Page 6: "PLS-DA (Partial least squares-discriminant analysis) can be regarded as a linear binary classifier." This is not true. PLS-DA can also be used for more than two classes.
9) As reported by many papers (see, e.g., Rinnan et al, 2009) SNV and MSC provide highly correlated corrections, so it does not make sense to apply them sequentially. By the way, the reference for SNV is duplicated (23 and 27)
10) My main source of criticism is that, especially considering the lack of really independent samples, as pointed out in 2) and 3), the validation strategy is not clear. There is no true model selection, since the results of the different pretreatments and of the different algorithms are presented and compared directly on the test set, but how model metaparameters were chosen and optimized is not reported. How was the number of latent variables in PLS-DA chosen? What was the kernel in SVM and how were the kernel and the slack parameters optimized? What was the architecture of the ANN? How many hidden neurons? What were the values of learning rate and momentum? How many neighbors in kNN? These are fundamental aspects which should be clarified, since the reliability of the reported results strictly depends on whether a proper model selection and model validation has been performed. In the present study, these aspects are rather obscure and should be clarified in possible resumbission
Author Response
Thank you very much for your valuable comments. Amendments have been made in the revised version of the manuscript, to address your concern. Please see the attached file. Thank you.

Reviewer 3 Report
Comments and Suggestions for Authors
The authors point out that application of a portable NIR spectrometer "to detect SARS-CoV-2 or the effect of the viral infection has rarely been reported in literature, partly due to the limitation of low resolution and the limited accuracy of such instruments". Therefore, they proposed to investigate if Artificial Intelligence and machine learning approach would allow to mitigate this deficiency.
1. I'm not an expert in machine learning procedures and application of AI for problem solving, but the connection between experimental conditions concerning the investigated samples, such as NIR spectra of a virus, concentration of the virus in the investigated samples and the interpretation of collected NIR spectroscopy data seems entirely superficial. Why the authors link the 1450 nm absorption peak with the presence of a virus?
2. The authors claim that they were able to solve the reproducibility problem of the investigated portable instrument by employing artificial intelligence, and they expect that the results can be even enhanced "if better computation resources are available". This may be redundant as the reproducibility error can be related to the applied methodology of collecting data. A rigorous analysis would have to be performed to determine if the employed instrument was the significant source of that error.
3. The authors conclude that the resolution of a portable NIR spectrometer has prevented them from demonstrating the usefulness of the proposed method for detecting a virus. It can be true. However, I suggest that they, first, validate the method with a high-resolution spectrometer and, second, identify a spectroscopic fingerprint of a virus for future investigation.
Author Response

(The authors gave the same response as above.)

Round 2
Reviewer 2 Report
Comments and Suggestions for Authors
The authors have taken a significant effort in addressing my comments to their previous submission and the quality of the manuscript has consequently improved. However, in my opinion, they still haven't fully grasped what my main source of criticism was and still is, which has to do to the way the authors validate their models.
I understand that the study should be considered as a proof of concept and that, for various reasons, the authors had to use samples which are not independent but should be considered as subsamples/aliquots of a single stock solution of viral species, but for those same reasons, a more conservative validation strategy should be enacted for the chemometric models.
Indeed, by looking at the newly introduced Figure 4 (by the way, authors should include a caption and correct some typos/mistakes in the text reported in the scheme), it is apparent how the modeling procedure involves several steps, each of which has a high potential to lead to overfitting.
On the other hand, as reported in the newly added section 2.5, for each algorithms, model parameters were optimized automatically based, presumably, since it is not explicitly reported, on the validation set. In this way, the validation set is never independent and cannot be considered for model comparison. Moreover, the results with random 70%/30% splitting shouldn't be reported as they suffer even more for the same lack of sample set independence discussed before.
A correct procedure should be to use e.g., A and B for model building, C for model selection (meaning that all the possible technique, model parameters, preprocessing are compared on this set) and only the final optimal model is tested on D. This would at least guarantee a fair validation, which is still lacking in the present revised approach.
Minor (?) comment:
In point (1) of my previous report, the main problem was not the use of PBS, but the relatively high concentration of the virus.
Reviewer 3 Report
Comments and Suggestions for Authors
Validation of the method with a high-resolution spectrometer is required before relying heavily on AI generated data.